# Extracellular Vesicle-Based Drug Delivery Systems in Cancer Therapy

**DOI:** 10.3390/ijms26104835

**Published:** 2025-05-19

**Authors:** Jiahao Wu, Zhesi Jin, Tingyu Fu, Yu Qian, Xinyue Bian, Xu Zhang, Jiahui Zhang

**Affiliations:** Jiangsu Key Laboratory of Medical Science and Laboratory Medicine, School of Medicine, Jiangsu University, Zhenjiang 212013, China; 3211401171@stmail.ujs.edu.cn (J.W.); jinzhesi@126.com (Z.J.); 18367545800@163.com (T.F.); qy_0411@163.com (Y.Q.); 18053472937@163.com (X.B.)

**Keywords:** extracellular vesicles, drug delivery, cancer therapy, biotherapy

## Abstract

Extracellular vesicles (EVs) are lipid bilayer-enclosed particles secreted by cells and ubiquitously present in various biofluids. They not only mediate intercellular communication but also serve as promising drug carriers that are capable of delivering therapeutic agents to target cells through their inherent physicochemical properties. In this review, we summarized the recent advances in EV isolation techniques and innovative drug-loading strategies. Furthermore, we emphasized the distinct advantages and therapeutic applications of EVs derived from different cellular sources in cancer treatment. Finally, we critically evaluated the ongoing clinical trials utilizing EVs for drug delivery and systematically assessed both the opportunities and challenges associated with implementing EV-based drug delivery systems in cancer therapy.

## 1. Introduction

Cancer remains one of the most formidable threats to human health and life. According to global statistics from 2022, there were nearly 20 million new cancer cases and 9.7 million cancer-related deaths worldwide that year [1]. Data from 2021 further revealed that cancer accounted for 14.57% of total deaths and 8.8% of total disability-adjusted life years (DALYs) globally across both sexes. The age-standardized incidence rate (ASIR) and age-standardized death rate (ASDR) were 790.33 and 116.49 per 100,000 population, respectively [2]. These metrics underscore the pervasive burden of cancer as a critical public health challenge. Despite significant progress in clinical interventions, such as chemotherapy, radiotherapy, and targeted therapies, these approaches are frequently limited by severe adverse effects and drug resistance [3,4]. For instance, chemotherapeutic agents indiscriminately damage both malignant and healthy cells, leading to complications like alopecia, myelosuppression, and gastrointestinal toxicity [5].

Nanoparticle-based drug delivery systems have emerged as promising therapeutic platforms due to their ability to enhance targeting precision while minimizing off-target toxicity [6]. However, conventional synthetic nanomaterials (e.g., liposomes, polymeric nanoparticles) face persistent challenges in biocompatibility, immunogenicity, and premature clearance by the reticuloendothelial system [7]. Extracellular vesicles (EVs), naturally derived nanoscale carriers, have revolutionized drug delivery research by leveraging their intrinsic biocompatibility, low immunogenicity, and inherent targeting capabilities [8,9].

EVs are phospholipid-bilayer nanoparticles actively secreted by cells and are ubiquitously detected in physiological fluids such as blood, urine, and saliva [10,11]. They serve not only as mediators of intercellular communication but also as promising drug carriers capable of targeted therapeutic delivery [12,13]. Compared to synthetic nanomaterials, EVs exhibit superior clinical advantages: their autologous or allogeneic cellular origin confers high biocompatibility, enabling evasion of immune recognition and clearance [14]; they demonstrate unique biological barrier penetration capabilities, including traversing the blood–brain barrier and tumor stroma [15]; and their surface molecules facilitate specific binding to target cells for precision targeting [16] (Figure 1).

This systematic review employs evidence-based methodology to investigate the transformative potential of EVs as next-generation drug delivery platforms in cancer therapeutics, with particular emphasis on three critical aspects: (1) the source-dependent therapeutic advantages of EV-based drug carriers, (2) current application strategies in cancer treatments, and (3) recent advances in clinical translation. The analysis integrates mechanistic insights with translational research progress to provide a critical perspective on EV-based cancer therapeutics.

## 2. Extracellular Vesicles

### 2.1. Biological Characteristics of Extracellular Vesicles

The intrinsic biological properties of EVs underpin their efficacy as drug delivery systems. Based on biogenesis mechanisms and dimensional characteristics, EVs are primarily classified into three subtypes: (1) exosomes (30–150 nm diameter), derived from multivesicular body (MVB)-plasma membrane fusion [17]; (2) microvesicles (100–1000 nm), generated through the direct outward budding of the plasma membrane [18]; and (3) apoptotic bodies (>500 nm), which emerge during programmed cell death through cytoplasmic blebbing [19,20].

EVs exhibit a complex molecular composition, comprising proteins, lipids, and nucleic acids (e.g., mRNA, miRNA, DNA). These cargo molecules not only mirror the phenotypic signature of their parental cells but also play pivotal roles in intercellular communication [21]. For instance, tumor-derived EVs act as key mediators in the tumor microenvironment, facilitating immune evasion and promoting metastatic processes through the horizontal transfer of oncogenic molecules [22].

The release of EVs is dynamically regulated by cellular stressors (e.g., hypoxia, oxidative stress), signaling pathways (e.g., Rab GTPase-dependent trafficking), and microenvironmental alterations (e.g., pH changes) [23]. Hypoxic conditions have been shown to upregulate exosome secretion from tumor cells, enriching their cargo with pro-angiogenic factors (e.g., VEGF, HIF-1α) that drive tumor progression [24]. Similarly, acidic microenvironments enhance both the release and uptake of tumor-derived exosomes while inducing cargo remodeling to favor pro-invasive phenotypes [25]. Elucidating these regulatory mechanisms is critical for optimizing EV-based drug delivery platforms in oncology.

### 2.2. EV Isolation Methodologies

Extracellular vesicle (EV) isolation employs four principal methodologies: ultracentrifugation, size-exclusion chromatography (SEC), affinity-based enrichment, and microfluidic technologies [26]. While ultracentrifugation remains the predominant method, it is limited by suboptimal efficiency and purity. Emerging approaches, such as affinity-based enrichment and microfluidics, demonstrate enhanced specificity and yield, addressing critical challenges in EV isolation [27].

Ultracentrifugation, the current gold-standard method, exploits the differential sedimentation coefficients of EVs versus other cellular components under centrifugal force. The most widely employed modes of ultracentrifugation include differential centrifugation and density gradient centrifugation. Differential centrifugation involves sequential centrifugation steps at progressively increasing centrifugal forces/speeds to separate heterogeneous particles, including extracellular vesicles (EVs), based on their size and density. Density gradient centrifugation utilizes reagents such as sucrose, deuterium oxide, and/or iodixanol (OptiPrep) to establish a continuous density gradient or discontinuous cushion. This enables the refined separation of EVs and their subpopulations by exploiting differences in buoyant density [28]. Despite its standardized protocols, this method necessitates expensive ultracentrifugation equipment and prolonged processing times (>10 h) and risks mechanical shear damage to EVs, potentially compromising their bioactivity and downstream functional applications [29].

Size-exclusion chromatography (SEC), also known as gel filtration chromatography, is a chromatographic method that separates particles into distinct fractions during elution based on the relative relationship between the pore size of the stationary phase packing material and the Stokes radius (i.e., the apparent molecular dimensions in solution) of the analytes. SEC is widely employed for EV isolation from complex biofluids. Although SEC effectively depletes high-abundance proteins like albumin, traditional workflows require multiple fractionation steps with tandem chromatographic columns. To address this, Lin et al. optimized a simplified binary SEC protocol using CL-6B resin, an expanded column bed volume (20 mL), and large elution volumes (8 mL), enabling the efficient separation of EVs from protein contaminants in just two elution steps [30]. Comparative studies by Yang et al. established that SEC-derived EVs exhibit superior RNA integrity for miRNA/mRNA sequencing and higher concordance with ideal FPKM (fragments per kilobase million) values in plasma omics analyses, positioning SEC as the optimal strategy for EV-based biomarker discovery [31]. Further advancements include Kapoor et al.’s size-exclusion fast protein liquid chromatography (SE-FPLC), achieving high EV recovery (88.47%) within 20 min while effectively removing lipoprotein contaminants from human/mouse serum and cell-derived EVs [32]. Liu et al. integrated SEC with tangential flow filtration to massively produce functionally distinct sEV subpopulations (S1-sEVs and S2-sEVs) from human umbilical cord mesenchymal stem cells (hUC-MSCs). S1-sEVs (CD9+/HRS+/GPC1+) showed potent immunomodulatory effects, whereas S2-sEVs (CD63+/FLOT1/2+) enhanced angiogenesis and proliferation in disease models [33].

Affinity-based enrichment is a methodology that employs molecules exhibiting high binding affinity to surface markers on EVs, either through facilitating the binding and enhancing the sedimentation coefficient of EVs or by utilizing magnetic beads to achieve EV enrichment. Immunoaffinity capture using magnetic beads functionalized with EV surface markers (CD63/CD9/CD81) achieves high specificity but requires harsh elution conditions (e.g., low pH) that may damage EVs. Brambilla et al. innovated a DNA-directed antibody immobilization system, enhancing capture efficiency via flexible DNA linkers and enabling gentle EV release using DNase I endonuclease [34]. Di et al. developed an automated Fe_3_O_4_@TiO_2_ bead-based platform for simultaneous EV enrichment and miRNA extraction within 30 min, outperforming TRIzol and commercial kits in RNA yield [35]. Wang et al. engineered CD63-antibody-conjugated cellulose nanofibers, achieving 86.4% EV capture efficiency through enhanced surface-area interactions [36].

Microfluidic technology enables the efficient separation, purification, or active generation of EVs through precisely engineered microscale channels and fluid control mechanisms, leveraging physical effects such as laminar flow, shear stress, or electric fields. Bajo-Santos et al. designed a cyclic olefin copolymer (COC)-based asymmetrical flow field-flow fractionation (A4F) device for continuous-flow EV separation, scalable for industrial/clinical production [37]. Loeng et al. fabricated a thiol-ene polymer-based microfluidic SEC (μSEC) platform with rapid flow-switching capabilities, enabling automated integration with downstream EV analytics [38]. Xin et al. engineered a recyclable boronate organic framework (BOF)-coated microfluidic chip, leveraging ROS-responsive phenylboronic ester crosslinking to achieve size-tunable (10–300 nm pores) EV isolation with reduced cost and technical demands compared to ultracentrifugation [39].

Mechanical extrusion generates EV-mimetic cell-derived nanovesicles (CNVs) with 10-fold higher yield and batch consistency than natural EVs. Lau et al.’s proprietary BioDrone™ platform produces GMP-compliant CNVs from umbilical cord MSCs (UCMSCs) that retain parental cell bioactivity, facilitating scalable EV-based therapeutic development [40].

Ultrasound-induced EV modulation exhibits intensity-dependent effects on EV secretion and functionality. Low-intensity pulsed ultrasound (LIPUS) enhances BMSC-EV yield and anti-inflammatory potency via MAPK/ERK and PI3K/Akt pathway activation [41]. Conversely, pulsed focused ultrasound (pFUS) upregulates regenerative pathways (eNOS/SIRT3) in MSC-EVs, augmenting their efficacy in acute kidney injury models [42]. Zmievskaya et al. demonstrated that ultrasound outperforms chemical induction (e.g., cytochalasin B) in generating T-cell EVs with enriched membrane proteins and therapeutic payload capacity, offering novel strategies to overcome CAR-T resistance in immunosuppressive tumors [43].

Beyond conventional vesicle preparation methods, the development of giant unilamellar vesicles (GUVs) has garnered increasing attention in recent years. Among existing techniques, electroformation is the most widely utilized approach [44]. This method employs an alternating electric field with specific frequency and intensity to induce asymmetric mechanical stress on phospholipid bilayers by leveraging the electrical properties of lipid headgroups, thereby promoting membrane bending, budding, and eventual vesicle closure [45]. However, traditional electroformation protocols often compromise GUV quality due to residual dried lipid films, which reduce the cholesterol concentration within the bilayer. To address this limitation, Boban et al. proposed an optimized electroformation strategy, combining rapid solvent exchange, plasma cleaning, and spin-coating techniques, to reproducibly generate GUVs from hydrated lipid films. Compared to conventional protocols, this method yields vesicles of comparable size but superior structural integrity [46]. In contrast, Waeterschoot et al. introduced a novel approach leveraging fluorinated silica nanoparticles (FNPs) to destabilize lipid-based nanosystems, enabling GUV formation under diverse buffer conditions while preventing the leakage of encapsulated components into the oil phase. A simple centrifugation step efficiently releases emulsion-trapped GUVs without requiring destabilizing chemicals. Subsequent evaluations of lipid lateral diffusion and GUV unilamellarity confirmed performance parity with electroformation-derived vesicles [47]. Furthermore, Ernits et al. established a microfluidics-based method for the direct synthesis of monodisperse GUVs (~100 μm diameter). Experimental results demonstrated a vesicle half-life of 61 ± 2 h, coupled with the efficient release of small dye molecules without significant membrane disruption. This technique reduces both the production time and cost compared to prior methodologies [48].

### 2.3. Drug-Loading Strategies

The current strategies for drug loading into extracellular vesicles (EVs) are classified as pre-loading during vesicle biogenesis and post-loading via isolation-coupled techniques (Figure 2).

#### 2.3.1. Pre-Loading

Pre-loading entails genetically or molecularly engineering parental cells to produce EVs inherently loaded with therapeutic cargo. This method leverages the cellular machinery to encapsulate functional molecules (e.g., nucleic acids, proteins) during EV biogenesis. Current pre-loading approaches primarily include co-incubation and transfection [49].

Co-incubation entails exposing parental cells to therapeutic agents under controlled conditions, allowing passive drug uptake through interactions with the phospholipid bilayer of the cell membrane. Subsequently, these molecules are packaged into EVs secreted by the cells. While straightforward, this method suffers from low loading efficiency, which varies significantly depending on drug properties (e.g., hydrophobicity, molecular weight), drug concentration, and the biological characteristics of parental cells [50].

Transfection, in contrast, employs genetic engineering to overexpress therapeutic molecules (e.g., miRNAs, siRNAs, or proteins) within parental cells, enabling the active incorporation of these molecules into secreted EVs. Although this approach offers higher reproducibility, challenges such as low transfection efficiency and cell viability-dependent outcomes persist. Recent studies demonstrate that engineering leukocyte-derived EVs with retroviral-like mRNA-packaging capsids enhances mRNA encapsulation efficiency and neuronal endocytosis. These autologous EVs exhibit immune inertness and improve neuron-specific mRNA delivery in murine models of chronic neuroinflammation [51].

#### 2.3.2. Post-Loading

Post-loading strategies involve exogenous cargo integration into isolated extracellular vesicles (EVs) through membrane permeabilization or physicochemical loading techniques. These approaches enable the encapsulation of small-molecule drugs, therapeutic proteins, and functional nanomaterials into EVs post-isolation, offering enhanced technical accessibility compared to parental cell-based engineering methods. Standardized protocols include electroporation, sonication, extrusion, and freeze–thaw cycles [52].

Electroporation employs pulsed electric fields to induce transient membrane destabilization in extracellular vesicles, generating hydrophilic membrane apertures that enable the concentration gradient-driven transport of diverse therapeutic cargo, including small-molecule pharmaceuticals, nucleic acid payloads, and protein biologics, into the vesicular compartment. Loading efficiency depends on multiple parameters, including vesicle density, the drug-to-EV ratio, buffer composition, pulse capacitance, and electric field intensity parameters [53]. Vesicle size critically determines the DNA loading capacity, with microvesicles demonstrating superior linear and plasmid DNA encapsulation compared to exosome-like EVs [54]. Optimizing electroporation parameters enhances protocol efficacy. Torabi et al. demonstrated a refined approach, combining magnetic bead sorting with electroporation for efficient miRNA loading into CD81+ EVs. This method achieved comparable miRNA encapsulation efficiency to conventional bulk electroporation while enriching CD81+ EV populations and enabling flexible parameter optimization, thereby advancing therapeutic RNA delivery systems with improved specificity and reduced cytotoxicity [55]. Additionally, studies demonstrate that linear DNA fragments (<1000 bp) exhibit higher association efficiency with EVs via electroporation than larger plasmids, with hundreds of DNA molecules per vesicle achievable [54].

Sonication employs ultrasonic waves to reversibly alter EV membrane permeability or induce transient pores, enabling drug encapsulation. For instance, Yuan et al. employed an ultrasound–microfluidics approach to achieve a 31.1% loading efficiency for siRNA targeting heat-shock protein 47 (siHSP47) into exosomes. The resulting exosome–siHSP47 complexes effectively penetrated collagen barriers and suppressed HSP47 expression in activated fibroblasts in vitro [56]. Methodological comparative analysis of EV-loading strategies—including sonication, electroporation, freeze–thaw cycles, and passive incubation—demonstrates that ultrasonic processing yields superior drug encapsulation efficiency while compromising vesicle yield. Conversely, electroporation maintains vesicle quantities at the expense of lower payload efficiency, highlighting a critical trade-off between cargo-loading capacity and EV integrity preservation [57]. Similarly, Jiang et al. demonstrated that the sonication-based loading of anthocyanins (ACN) into milk-derived EVs (MEVs) produced stable, nanosized MEV-ACN complexes with enhanced stability under simulated gastrointestinal conditions [58].

Freeze–thaw cycles leverage the physical disruption of the EV lipid bilayer to promote drug entrapment. Hettich et al. systematically compared hydrophilic compound loading via saponin permeabilization, sonication, membrane fusion, freeze–thaw cycles, and osmotic shock. Osmotic shock and freeze–thaw methods outperformed others, achieving superior drug encapsulation while preserving EV structural and functional properties [59]. This highlights the importance of balancing loading efficiency with EV integrity for therapeutic applications.

Emerging post-loading strategies have recently demonstrated remarkable improvements in efficiency and versatility. For example, Lee et al. developed a tension-controlled (TC) loading method that rapidly incorporates diverse cargo (e.g., doxorubicin, ssDNA, miRNA) into EVs. TC exhibited 4.3- to 7.2-fold higher loading rates than sonication or extrusion, with reduced heterogeneity and enhanced therapeutic efficacy in miRNA-497- or doxorubicin-loaded EVs [60]. Similarly, Zhang et al. developed a chirality-engineered graphene quantum dot (GQD)-EV platform that achieves exogenous cargo-agnostic loading through structural complementarity with EV lipid bilayers. Both hydrophobic small-molecule drugs (e.g., doxorubicin) and hydrophilic biologics (e.g., siRNA) can be efficiently functionalized onto GQDs via π-π stacking and van der Waals interactions. Experimental validation revealed loading efficiencies exceeding 60% for both payload classes (66.3% and 64.1%, respectively), demonstrating significant superiority over conventional EV-loading techniques [61]. Another innovative method, esterase-responsive active loading (EAL), leverages transmembrane pH gradients to encapsulate ferulic acid derivatives into EVs with 5- to 6-fold higher efficiency than passive methods, enabling sustained release and minimized cytotoxicity [62].

Simplified protocols for EV loading are also emerging. Kronstadt et al. introduced a pH gradient-based method for loading negatively charged nucleic acids into EVs, which requires no specialized equipment and utilizes standard laboratory reagents [63]. Bao et al. demonstrated that boron clusters enhance doxorubicin loading into exosomes by forming supramolecular complexes, which inhibit P-glycoprotein-mediated drug efflux and overcome chemoresistance in breast cancer models [64]. Furthermore, Zhang et al. utilized cell-penetrating peptides (CPPs) to boost doxorubicin encapsulation efficiency to 37.18%—2.5-fold higher than passive incubation—while maintaining operational simplicity [65].

Collectively, these advancements in post-loading strategies underscore the potential of extracellular vesicle-mediated drug delivery platforms for enhanced therapeutic precision, scalability, and clinical applicability.

## 3. Advantages of Extracellular Vesicle-Based Drug Delivery Systems from Diverse Cellular Origins in Cancer Therapy

In recent years, studies have found that extracellular vesicles (EVs) from different sources have different advantages as drug delivery systems in cancer treatment. For example, mammalian cell-derived extracellular vesicles have advantages such as tumor targeting and easy penetration of the blood-brain barrier, bacterial derived extracellular vesicles have advantages such as gastrointestinal stability and natural anti-tumor effects, while plant and milk derived extracellular vesicles have natural advantages such as wide source and oral administration (Figure 3).

### 3.1. Mammalian Cell-Derived EVs

#### 3.1.1. Mesenchymal Stem Cells (MSCs)

A growing body of experimental studies has recently demonstrated that mesenchymal stem cell (MSC)-derived extracellular vesicles (EVs) represent a promising drug delivery system for anticancer therapy due to their biocompatibility, ability to traverse biological barriers, and intrinsic targeting capabilities [66]. Research has shown that MSC-derived EVs can efficiently deliver chemotherapeutic agents, siRNA, and miRNA to tumor cells, enhancing therapeutic efficacy while minimizing off-target effects [67]. For instance, Liu et al. reported that MSC-derived EVs penetrate tumor tissues and successfully deliver miR-138-5p, effectively suppressing the growth of xenograft tumors in vivo [68]. Additionally, studies have revealed that the chemotactic properties of MSC-derived EVs toward osteosarcoma cells via the SDF1-CXCR4 axis enhance their targeting specificity. This mechanism enables the precise delivery of doxorubicin to osteosarcoma tissues, amplifying tumor toxicity while reducing cardiotoxicity [69]. Notably, MSC-derived EVs have also demonstrated an exceptional capacity to cross the blood–brain barrier (BBB). In a study by Song et al., rapamycin-loaded EVs derived from MSCs (EV-Rapa) successfully delivered the drug to the brain, penetrating GL261 cells to exert therapeutic effects. Furthermore, the acidic tumor microenvironment was shown to facilitate drug release from EVs [70].

Emerging evidence highlights the intrinsic antitumor potential of MSC-derived EVs. For example, Zhou et al. investigated the effects of human placental MSC-derived EVs (hPMSC-EVs) on the malignant behavior of 4T1 murine breast cancer models, both in vitro and in vivo. Their findings indicated that hPMSC-EVs suppress tumor proliferation and migration by indirect anti-angiogenic mechanisms, thereby inhibiting breast cancer progression [71]. Deeper mechanistic insights reveal that MSC-derived EVs are enriched with miRNAs possessing anticancer properties. Wu et al. identified miR-13896 in MSC-derived EVs, which targets and downregulates ATG2A-mediated autophagy pathways to inhibit gastric cancer cell growth and metastasis [72]. Similarly, Cui et al. demonstrated that MSC-derived EVs carrying miR-486-5p suppress colorectal cancer cell proliferation and metastasis by targeting NEK2, thereby inhibiting glycolysis and stemness in these cells [73].

#### 3.1.2. Immune Cells

Immune cell-derived extracellular vesicles (EVs) have emerged as promising candidates for targeted nanotherapeutics in oncology, capitalizing on their inherent biomimetic properties, tissue-specific tropism, and immunomodulatory functions to achieve enhanced therapeutic indices with reduced systemic toxicity [74,75]. EVs sourced from immune cells such as NK cells, T cells, and macrophages not only enhance the stability and bioavailability of therapeutic compounds but also activate antitumor immune responses through the delivery of immunoregulatory molecules like cytokines and antigens. For instance, Zhang et al. reported that cytokine-activated CD8+ T cell-derived EVs (caCD8-EVs) exhibit robust cytotoxicity against diverse cancer cells in vitro, mediated by cytotoxic proteins (granzyme B and perforin) and interferon gamma (IFNγ), synergistically inducing tumor cell apoptosis [76]. Similarly, Kim et al. demonstrated that NK cell-derived exosomes inhibit serine/threonine kinase phosphorylation in Hep3B hepatocellular carcinoma cells, promoting apoptotic marker activation and achieving targeted accumulation in both orthotopic and subcutaneous HCC mouse models [77]. Shi et al. further highlighted the superiority of memory-like NK cell-derived small EVs (mNK-sEVs), which suppress tumor growth more effectively than conventional NK-EVs (conNK-sEVs) via caspase-dependent apoptotic pathways, with enhanced pharmacokinetic retention in xenograft tumors [78]. Macrophage-derived EVs, as shown by Haney et al., display their natural tumor tropism to deliver chemotherapeutics and reprogram tumor-associated macrophages (TAMs), thereby amplifying antitumor immunity [79]. Additionally, Gambardella et al. revealed that IL-33-primed eosinophil-derived EVs upregulate cyclin-dependent kinase inhibitor (CDKI) genes, inducing G0/G1 cell cycle arrest and suppressing tumor spheroid formation, which underscores their role in halting cancer proliferation [80]. These investigations collectively elucidate the versatility of immune cell-derived EVs in integrating targeted drug delivery with immune activation for precision oncology.

#### 3.1.3. Tumor Cells

Tumor cell-derived extracellular vesicles (EVs) exhibit structural and compositional homology with their parental cell membranes, conferring high biocompatibility and reduced immune rejection while enabling precise drug delivery through surface-displayed tumor-specific proteins and receptors [81]. These intrinsic targeting properties minimize off-tissue toxicity by facilitating selective binding to malignant cells. As exemplified by Jin et al., colorectal cancer (CRC)-derived PKH26-labeled exosomes are internalized more efficiently by SW1116 tumor cells than by EVs from mesenchymal stem cells (MSCs) or HepG2 hepatoma cells, with fluorescence imaging revealing robust intratumoral accumulation [82]. Similarly, Sulthana et al. demonstrated that 4T1 breast cancer-derived EVs preferentially accumulate in metastatic organs such as the liver, spleen, and lungs, correlating with in vivo imaging evidence of pulmonary metastasis [83]. Sancho-Albero et al. further corroborated this tissue specificity, showing that melanoma-derived B16-BL6 EVs selectively target lung tissues and are preferentially internalized by tumors in dual xenograft models [84]. Significantly, Vincenti et al. demonstrated that canine glioma-secreted EVs cross the blood–brain barrier (BBB) and exhibit preferential accumulation in intracranial tumors, underscoring their therapeutic promise for CNS-targeted drug delivery [85].

Beyond their role in targeted drug transport, tumor-derived EVs exhibit intrinsic antitumor activity. Wang et al. demonstrated that EVs from breast, pancreatic, and colorectal cancers reprogram immunosuppressive M2 macrophages toward the pro-inflammatory M1 phenotype, reactivating antitumor immunity [86]. These EVs also carry tumor-associated antigens (TAAs), which prime adaptive immune responses to enhance therapeutic efficacy [83]. Furthermore, tumor-derived EVs inherit elevated levels of 70 kDa heat-shock proteins (Hsp70s), upregulated during endoplasmic reticulum (ER) stress, which serve dual functions as endogenous adjuvants and lymphatic targeting enhancers. The subcutaneous administration of tumor EVs promotes their migration to lymph nodes (LNs), where rapid dendritic cell (DC) internalization synchronizes the delivery of TAAs and Hsp70s. This coordinated action robustly activates DC-mediated antigen presentation, eliciting robust oncoimmunological activation while inducing durable immunological imprinting [87]. Such multifaceted functionality, combining precision drug delivery with immune activation, positions tumor-derived EVs as multifunctional platforms for next-generation cancer therapeutics.

### 3.2. Bacteria-Derived Extracellular Vesicles

Bacteria-derived extracellular vesicles (EVs) have emerged as promising platforms for cancer drug delivery, leveraging their innate immunogenicity and intrinsic antitumor properties [88]. For instance, EVs from the probiotic *Lactobacillus reuteri* (REVs) exhibit exceptional stability in the gastrointestinal tract, exerting antitumor effects through the modulation of apoptotic signaling pathways. When combined with chemotherapeutic agents, REVs enhance tumor ablation efficacy and induce immunogenic cell death, demonstrating their dual therapeutic potential [89]. Similarly, *Lactobacillus paracasei*-derived EVs (LpEVs) curb colorectal cancer cell proliferation by suppressing HIF-1α-mediated glycolysis, thereby disrupting the energy metabolism essential for tumor growth [90]. Parallel findings by Zhang et al. revealed that *Lactobacillus plantarum*-derived EVs (LEVs) suppress colorectal cancer progression via SIRT5 downregulation, which modulates p53 desuccinylation to attenuate glycolysis and proliferation [91]. Beyond their direct antitumor effects, *Bifidobacterium longum*-derived EVs reverse carboplatin resistance in ovarian cancer cells by promoting p53 Ser15 phosphorylation, sensitizing tumors to chemotherapy and indirectly enhancing therapeutic outcomes [92].

Despite their therapeutic potential, the scalable production of bacterial EVs remains a critical challenge for clinical translation. To address this, Won et al. developed a robust manufacturing process for *Escherichia coli*-derived outer membrane vesicles (OMVs), which exhibit potent immunostimulatory properties. These OMVs enhance the intratumoral infiltration and activation of CD8+ T cells, particularly those expressing high levels of TCF-1 and PD-1, thereby amplifying antigen-specific antitumor immunity. Notably, *E. coli* OMVs synergize with anti-PD-1 checkpoint inhibitors by promoting the recruitment of stem-like CD8+ T cells into the tumor microenvironment, achieving significant tumor growth suppression in preclinical models [93]. Furthermore, Xiao et al. demonstrated that OMVs reprogram tumor-associated macrophages toward the pro-inflammatory M1 phenotype, stimulating the release of cytokines such as TNF-α and IL-12 to establish an immunologically hostile milieu for cancer progression [94]. Collectively, these studies elucidate the bifunctional capacity of bacterial EVs as both targeted drug carriers and immunomodulators, bridging the gap between microbial biology and precision oncology.

### 3.3. Plant-Derived Extracellular Vesicles

Plant-derived extracellular vesicles (PDEs), nanoscale vesicles secreted by multivesicular bodies, play pivotal roles in gene regulation, intercellular communication, and pathogen defense while exhibiting low cytotoxicity toward healthy tissues and tumor-targeting specificity through unique endocytic mechanisms [95]. These attributes position PDEs as promising candidates for next-generation biotherapeutics and drug delivery platforms. Recent studies highlight the anticancer potential of plant EVs, exemplified by Perilla frutescens-derived EVs, which suppress the proliferation, migration, and invasion of breast cancer cells via caveolin-1-dependent pathways, sparing normal cells due to their selective uptake [96]. Similarly, celery-derived EVs (CDEVs) attenuate the phospho-STAT3/PD-L1 signaling axis in pulmonary carcinoma cells, effectively decoupling PD-1/PD-L1 immune checkpoints to abrogate T-cell depletion and reinvigorate oncoimmunological surveillance [97]. Platycodon grandiflorus-derived EVs (PGEVs) demonstrate remarkable stability under simulated gastrointestinal conditions, activating oxidative stress-triggered programmed cell death in neoplastic cells concurrent with reprogramming tumor-associated macrophages (TAMs) into immunostimulatory M1 subsets, thereby amplifying Th1-type cytokine production. In vivo, PGEVs accumulate preferentially in 4T1 tumors, modulating gut microbiota and systemic antitumor immunity for enhanced therapeutic efficacy [98]. Garlic-derived EVs further exemplify tumor-selective cytotoxicity by activating caspase-dependent apoptosis without affecting healthy cells [99]. Additionally, a study by Xu et al. demonstrated that orally administered garlic-derived nanoparticles (GNPs) effectively activate intestinal IFN-γ-producing γδ T cells, promote their migration to distant tumor sites, and remodel the immunosuppressive tumor microenvironment, thereby suppressing tumor progression [100].

Despite their therapeutic promise, the scalable production of plant EVs remains challenging. To resolve this, Ou et al. established a Catharanthus roseus cell culture system, yielding exosome-like nanovesicles with tripled production efficiency compared to traditional methods while retaining bioactive lipids, proteins, and RNA critical for their therapeutic function [101]. These plant-derived exosome-like nanoparticles exhibit exceptional stability and biocompatibility, facilitating oral delivery and the gut-targeted modulation of inflammatory and oncogenic pathways via plant miRNA activity [102]. Citron-derived extracellular nanovesicles (CLENs) further suppress cancer progression by inhibiting PI3K/AKT and MAPK/ERK signaling, thereby blocking proliferation and metastasis [103]. Ginger-derived EVs (GEVs), enriched with cytotoxic gingerols and shogaols, induce apoptosis through p53 pathway activation and cell cycle arrest, underscoring their dual role as cytotoxic agents and drug carriers [104]. Notably, basil-derived EVs exhibit potent pro-apoptotic effects in pancreatic cancer models, leveraging their inherent anti-inflammatory and anticancer properties [105].

Collectively, plant-derived EVs combine biocompatibility, tumor-targeting precision, and multimodal antitumor mechanisms—ranging from immune modulation to direct cytotoxicity—offering a versatile platform for drug delivery and cancer therapy. Their integration of bioactive phytochemicals and endogenous nucleic acids positions them at the forefront of natural nanomedicine development.

### 3.4. Milk-Derived Extracellular Vesicles

Milk-derived extracellular vesicles (EVs) demonstrate unique advantages as drug delivery systems, including high production yields, oral bioavailability, and inherent therapeutic benefits, compared to synthetic nanocarriers [106,107]. González-Sarrías et al. elucidated that milk-derived EVs enhance the bioavailability and anticancer activity of polyphenols by shielding these compounds from metabolic degradation and delivering them to mammary tissues via clathrin-mediated endocytosis. This mechanism circumvents ATP-binding cassette transporter-mediated chemoresistance in cancer cells, enabling rapid and potent anticancer effects at therapeutically effective concentrations [108]. Reif et al. further elucidated the dual role of milk-derived EVs (MDEs): they promote intestinal epithelial proliferation and barrier repair in healthy colon organoids by upregulating β-catenin, cyclin D1, and the proliferation marker Ki67 while simultaneously suppressing colorectal cancer progression through the downregulation of these pathways, thereby mitigating malignancy risk [109].

The drug delivery efficiency of milk-derived EVs varies significantly across species. Comparative studies isolating and characterizing EVs from cow, buffalo, and goat milk revealed that goat milk EVs exhibit superior encapsulation efficiency and drug-loading capacity across multiple methodologies, positioning them as optimal carriers for hydrophobic therapeutics [110]. Notably, camel milk-derived EVs have emerged as a particularly promising platform. Camel milk EVs loaded with ARV-825, a proteolysis-targeting chimera, demonstrated enhanced sustained drug release kinetics, permeability, and systemic bioavailability compared to free drug formulations, with improved pharmacokinetic absorption profiles and plasma exposure [111]. These findings collectively highlight the potential of camel milk-derived EVs as high-performance delivery systems to overcome existing limitations in cancer therapy.

By integrating species-specific optimization, scalable production, and intrinsic biocompatibility, milk-derived EVs represent a transformative strategy for oral drug delivery, bridging the gap between natural nanocarriers and precision oncology (Table 1).

Beyond inherent natural advantages, the integration of extracellular vesicles (EVs) with novel nanomaterials holds significant potential to further enhance their functional performance. Among these advancements, the combination of EVs with magnetic nanomaterials has attracted considerable attention. This hybrid system retains the intrinsic biological properties of EVs while incorporating the magnetic characteristics of nanomaterials, offering unique advantages in tumor-targeting specificity and therapeutic efficacy, thereby positioning itself as a promising strategy for optimizing EV-based drug delivery systems [112]. For instance, Wang et al. engineered EV-mimetic nanovesicles (NVs) encapsulating iron oxide nanoparticles (IONPs) and β-Lapachone (Lapa). Experimental results demonstrated that the designed NV-IONP-Lapa exhibited enhanced tumor-targeting specificity, facilitated MR imaging, and improved tumor suppression without significant side effects [113].

## 4. Application of Extracellular Vesicle-Based Drug Delivery Systems in Cancer Therapy

Extracellular vesicles (EVs) serve as versatile drug delivery systems for chemotherapy, radiotherapy, gene therapy, immunotherapy, photothermal therapy, and cancer vaccines. Their biocompatibility and cargo-loading capacity enable the targeted delivery of therapeutics while minimizing systemic toxicity, offering a multifunctional platform for integrated cancer treatment strategies (Figure 4).

### 4.1. Chemotherapy Applications

#### 4.1.1. Anthracyclines

Doxorubicin (DOX), a first-line anthracycline chemotherapeutic agent for various malignancies, is limited by its severe cardiotoxicity. Recent studies highlight the cardioprotective potential of extracellular vesicles (EVs) derived from medicinal plants. For instance, bitter melon-derived EVs attenuate doxorubicin-induced cardiotoxicity by suppressing ubiquitin-dependent SQSTM1/p62 degradation. This intervention reactivates the SQSTM1-KEAP1 complex assembly, enhances NFE2L2 nuclear translocation, and induces HMOX1 transcription, thereby improving cardiac function and myocardial architecture in preclinical models [114]. To address tumor resistance and enhance therapeutic efficacy, Shamshiripour et al. engineered dendritic cell-derived EVs co-loaded with VEGF-A siRNA and DOX. This dual-loading strategy not only reduced tumor angiogenesis but also synergistically induced apoptosis, outperforming bevacizumab (BV) monotherapy while leveraging the immunomodulatory benefits of DC-EVs [115].

Innovative targeting approaches further refine EV-based DOX delivery. Wiklander et al. developed an EV–antibody conjugate system by displaying tumor-specific antibodies on Fc-engineered EVs (Fc-EVs), enabling precise tumor targeting. Fc-EVs encapsulating DOX demonstrated enhanced therapeutic efficacy with reduced off-target toxicity compared to conventional chemotherapy [116]. Similarly, Xu et al. engineered Rhodiola rosea-derived EV-like nanovesicles (RELNs) modified with DSPE-PEG2000-pYEEIE (pYEEIE) for melanoma therapy. The pYEEIE-RELN-DOX system exhibited superior antiproliferative effects and tumor-targeting specificity over free DOX or unmodified RELN-DOX, underscoring the importance of surface functionalization in optimizing drug delivery [117]. Additionally, lipid-coated EVs derived from breast cancer cells loaded with DOX demonstrated broad cytotoxicity across molecular subtypes of breast cancer, enhanced antimigratory properties, and comparable cellular uptake to free DOX. Critically, these DOX-loaded EVs showed no histopathological damage to the spleen, liver, heart, bone marrow, or kidneys at the median lethal dose, confirming their favorable biosafety profile [118].

Collectively, these strategies—spanning natural EV cardioprotection, siRNA co-delivery, antibody-directed targeting, surface engineering, and tumor cell-derived EV platforms—provide innovative solutions to mitigate DOX-associated cardiotoxicity while enhancing antitumor efficacy, marking significant advances in precision oncology.

#### 4.1.2. Plant Alkaloids

Paclitaxel (PTX), a plant alkaloid-derived chemotherapeutic agent, is widely used in clinical oncology but limited by systemic toxicity due to nonspecific biodistribution, often causing multi-organ damage, particularly to the liver and kidneys. Recent advances in extracellular vesicle (EV)-based delivery systems have addressed these challenges while enhancing tumor targeting. Zheng et al. developed inhalable CAR-T cell-derived EVs surface-functionalized with anti-mesothelin (MSLN) scFv antibodies for the selective targeting of MSLN-overexpressing Lewis lung carcinoma (MSLN-LLC). In orthotopic pulmonary malignancy models, aerosolized PTX-loaded CAR-EVs (PTX@CAR-EVs) exhibited precision tumor localization, driving significant tumor regression, survival extension, and attenuated off-target effects versus free PTX administration [119]. Creeden et al. engineered bispecific EVs (ExoSmart) functionalized with RGD peptides and CD47p110-130, demonstrating selective αvβ3 integrin targeting in pancreatic ductal adenocarcinoma (PDAC). This dual-modality platform potentiated cellular internalization and PTX efficacy across experimental models, concurrent with CD47p110-130-driven SIRPα blockade in macrophages reducing hepatic and splenic clearance, thereby extending systemic EV persistence [120].

Innovative surface engineering strategies further optimize PTX delivery. Chen et al. functionalized milk-derived EVs (MEVs) using Streptomyces phospholipase D to anchor transferrin ligands, achieving the significantly enhanced cellular uptake and cytotoxicity of PTX-loaded MEVs in cancer models [121]. Talatapeh et al. developed large EVs (LEVs) loaded with PTX (LEVs-PTX), which induced tumor cell apoptosis while suppressing pro-survival autophagy pathways and upregulating mitophagy markers, synergistically inhibiting tumor growth [122]. For triple-negative breast cancer (TNBC), Bhullar et al. engineered CD44 aptamer-modified EVs co-loaded with PTX and survivin-targeting siRNA (SUR-siRNA). This system demonstrated tumor-specific targeting, minimized off-target effects, and achieved significant tumor suppression in xenograft models at reduced PTX doses, markedly lowering adverse effects [123].

Intriguingly, PTX itself may enhance EV targeting capabilities. Studies reveal that PTX loading into mesenchymal stem cell-derived EVs (MSC-EVs) upregulates stromal cell-derived factor-1 (SDF-1) expression, conferring selective tropism toward CXCR4/CXCR7-expressing tumors. Concurrently, PTX-loaded MSC-EVs exhibit enriched antitumor microRNA cargo, amplifying therapeutic efficacy. This bidirectional synergy positions MSC-EVs as a novel platform for tumor-targeted drug delivery systems (TDDS) [124].

Collectively, these EV-based strategies—spanning CAR-mediated targeting, dual-functional surface engineering, biomimetic functionalization, and PTX-driven tropism enhancement—address the dual challenges of PTX toxicity and tumor resistance, ushering in a new era of precision chemotherapy.

#### 4.1.3. Alkylating Agents

Platinum-based alkylating agents, such as cisplatin and carboplatin, are widely used in oncology for their potent antitumor efficacy but are limited by dose-dependent neurotoxicity and off-target organ damage. Emerging studies highlight the capacity of extracellular vesicle (EV)-mediated delivery systems to reduce treatment-related toxicities and improve therapeutic targeting precision. For instance, EVs derived from NRF2-overexpressing neural progenitor cells attenuated cisplatin-induced neurotoxicity via the activation of the NRF2/antioxidant response element (ARE) pathway, thereby suppressing oxidative stress in neurons and preserving their viability in preclinical models [125]. Nathani et al. enhanced platinum drug delivery through IL-15-stimulated natural killer cell-derived EVs encapsulating carboplatin (CBP). This approach reduced the IC50 of CBP by 2.3-fold in 2D and 3D lung cancer cultures compared to the free drug while significantly decreasing tumor volume in xenograft models without exacerbating systemic toxicity [126].

To address delivery inefficiencies in colorectal cancer, Chandler et al. engineered EVs functionalized with the photosensitizer porphyrin (C5SHU) for oxaliplatin delivery. The C5SHU-EV system not only enhanced oxaliplatin uptake and cytotoxicity in cancer cells but also enabled light-triggered drug release, synergistically improving therapeutic outcomes while minimizing off-target effects. Remarkably, this study validated the utility of laser ablation inductively coupled plasma mass spectrometry and mass spectrometry imaging for quantifying platinum drug-loading efficiency and spatial distribution within tumors, establishing a robust framework for EV-based drug evaluation [127].

These advances spanning neuroprotective EV engineering, immune cell-derived EV platforms, and light-responsive delivery systems collectively highlight the translational potential of EV-mediated strategies to overcome the limitations of conventional platinum chemotherapy, balancing efficacy with reduced toxicity in precision oncology.

#### 4.1.4. Antimetabolites

Gemcitabine (GEM), a widely used antimetabolite chemotherapeutic agent, is constrained by low delivery efficiency and systemic toxicity. Recent studies highlight the capacity of extracellular vesicle (EV)-mediated systems to address these limitations. For instance, human bone marrow mesenchymal stem cell (MSC)-derived EVs loaded with GEM (Exo-GEM) were shown to enhance apoptosis in pancreatic cancer cells while maintaining favorable biosafety profiles in preclinical models, offering a promising strategy to balance efficacy and toxicity [128]. Kim et al. further optimized pancreatic ductal adenocarcinoma (PDAC) therapy by developing exosome–liposome hybrid nanoparticles encapsulating a GEM prodrug. Leveraging their inherent tumor-targeting capacity and macropinocytosis-mediated uptake, these nanoparticles exhibited superior specificity for pancreatic cancer cells, significantly improving drug accumulation at tumor sites [129]. To expand targeting applications, Gaurav et al. engineered GE11 peptide-modified EVs derived from human umbilical vein endothelial cells (HUVEC-EVs) for vinorelbine delivery in lung cancer. This system achieved tumor-specific targeting via epidermal growth factor receptor (EGFR) recognition, demonstrating potent antitumor effects and reduced off-target toxicity in murine models [130].

Methotrexate (MTX), a cornerstone therapy for central nervous system lymphoma (CNSL), faces challenges due to poor blood–brain barrier (BBB) penetration and myelosuppression. Zhao et al. addressed this by engineering anti-CD19 lentivirus-modified EVs from human adipose-derived MSCs (CD19-Exo) to deliver MTX. These EVs efficiently crossed the BBB, selectively accumulated in CNSL lesions, and reversed MTX-induced bone marrow suppression without detectable organ toxicity, highlighting their dual role in enhancing CNS delivery and mitigating adverse effects [131].

Notably, EVs can reverse chemoresistance, a critical barrier in antimetabolite therapy. Broccoli-derived EVs (BEVs) loaded with 5-fluorouracil (5-FU) were shown to suppress proliferation, migration, and cell cycle progression in colorectal cancer HT-29 cells while inducing apoptosis through reactive oxygen species (ROS) generation and mitochondrial dysfunction. Mechanistically, BEVs inhibited hyperactivation of the PI3K/Akt/mTOR pathway, restoring 5-FU sensitivity and overcoming drug resistance in preclinical models [132]. These findings underscore the dual capacity of EV-based systems to enhance therapeutic efficacy and re-sensitize tumors to conventional antimetabolites, offering transformative potential in precision oncology.

### 4.2. Application in Radiotherapy

Radioresistance remains a critical challenge in cancer therapy. Emerging evidence highlights the potential of extracellular vesicles (EVs) to overcome this limitation and enhance radiotherapeutic efficacy. For example, Chen et al. developed a porous microneedle system encapsulating STING agonist MSA-2-loaded exosome-mimetic nanoparticles (EXO). Leveraging the microneedle’s tumor-penetrating capacity, EXO enabled the sustained release and deep accumulation of MSA-2 in residual tumor tissues. Upon exposure to ultra-high dose-rate (FLASH) irradiation, MSA-2 activated the STING pathway, elevating type I interferon levels, promoting dendritic cell (DC) maturation, and reprogramming the immunosuppressive tumor microenvironment (TME), thereby mitigating radioresistance and optimizing FLASH radiotherapy outcomes [133]. To address radioresistance systematically, Yang et al. identified staurosporine (STS) as a potent radiosensitizer through the high-throughput screening of natural compounds. However, given STS’s inherent cytotoxicity, they engineered an EV-based delivery system (EV-STS) that selectively targets ATP-binding cassette subfamily A member 1 (ABCA1) in tumors. EV-STS significantly enhanced radiosensitivity in subcutaneous tumor models while maintaining favorable biosafety, demonstrating a dual strategy to amplify radiotherapy efficacy and minimize off-target toxicity [134].

Further advancing radio-immunotherapy, researchers engineered an autologous EV-based nanoplatform (MnExo@cGAMP) co-loaded with manganese ions (Mn^2+^) and cyclic GMP-AMP (2′,3′-cGAMP). Upon tumor cell internalization, Mn^2+^ ions potentiated the binding of the stimulator of interferon genes (STING) to radiotherapy-induced double-stranded DNA (dsDNA), while 2′,3′-cGAMP acted as a secondary messenger to amplify cGAS-STING pathway activation. This cascade triggered robust type I interferon production, enhanced DC-mediated antigen presentation, and activated tumor-specific CD8^+^ T cells, effectively overcoming immunosuppression in melanoma and achieving synergistic radio-immunotherapeutic effects [135].

Collectively, these EV-based strategies—spanning STING pathway activation, targeted radiosensitizer delivery, and immunostimulatory nanoplatforms—address radioresistance through multifaceted mechanisms, offering novel avenues to enhance precision radiotherapy and synergize with antitumor immunity.

### 4.3. Application in Gene Therapy

Extracellular vesicles (EVs) have demonstrated potential as engineered nanocarriers for therapeutic nucleic acid delivery, specifically transporting siRNA, miRNA, and CRISPR-Cas9 payloads to precisely regulate oncogene expression networks. For instance, CRISPR/Cas9 systems engineered within EVs have demonstrated the precise editing of immune checkpoint genes, including PD-1 and CTLA-4, leading to improved therapeutic responses in immunomodulatory interventions [136].

#### 4.3.1. siRNA Delivery

siRNA therapy holds promise for cancer treatment by silencing disease-driving genes, yet its clinical translation is hindered by poor stability, inefficient delivery, and systemic toxicity. Recent advancements in EV-based delivery systems address these challenges effectively. For example, edible and non-cationic kiwifruit-derived EVs (KEVs) demonstrated a sevenfold higher safety threshold than conventional cationic liposomes. KEVs encapsulating STAT3-targeting siRNA (siSTAT3) demonstrated enhanced structural integrity, targeted molecular recognition, and selective cytotoxicity in NSCLC cell lines, achieving substantial tumor growth inhibition in xenograft models [137]. Similarly, Rabienezhad et al. isolated extracellular vesicles from tangerine juice (TNVs) and loaded them with DDHD1-siRNA via electroporation (13% loading efficiency). TNV-delivered siRNA achieved a 60% reduction in target gene expression in colorectal cancer cells, highlighting the potential of plant-derived EVs in gene therapy [138].

To enhance delivery efficiency, Jiang et al. engineered EGFR-targeted exosomes (exo-scFv) derived from HEK293T cells by displaying single-chain variable fragments (scFv). These exosomes efficiently crossed the blood–brain barrier (BBB) and delivered LPCAT1-siRNA (siLPCAT1) to brain metastasis sites, inhibiting the malignant progression of lung cancer without detectable toxicity [139]. Kim G. et al. further optimized siRNA delivery by integrating chiral graphene quantum dots and pH-responsive peptides into small EVs (sEVs). This modification increased cytosolic cargo delivery by 1.74-fold, achieving 73% gene silencing efficiency, a significant improvement over unmodified sEVs [140].

EVs also enable combinatorial gene therapy. Rahmani et al. developed anti-EGFRvIII antibody-modified mesenchymal stem cell (MSC)-derived EVs co-loaded with two apoptosis-inducing agents for glioblastoma (GBM) treatment. These engineered EVs selectively accumulated in tumors, inducing higher apoptosis rates than monotherapy [141]. Gong et al. engineered mesenchymal stem cell-derived small extracellular vesicles (sEVs) with urokinase plasminogen activator (uPA) surface modifications, co-encapsulating Src siRNA to generate uPA-sEVs-siSrc complexes. This system dual-targeted chemotherapy-induced senescent stromal cells and tumor cells, synergizing with doxorubicin to reduce the senescence burden and enhance therapeutic outcomes [142].

Innovative engineering strategies further expand EV utility in gene therapy. Kim et al. engineered a “Shock Wave-mediated Extracellular Vesicle Engineering Technology” (SWEET) to encapsulate KRASG12C-targeting siRNA in small bovine milk-derived EVs (sBMEVs). In NSCLC xenografts, SWEET-generated sBMEVs silenced oncogene expression and suppressed tumor growth effectively [143]. Additionally, Taghavi-Farahabadi et al. repolarized M2 macrophages to an M1 phenotype using M1 macrophage-derived exosomes co-loaded with siSIRPα and siSTAT6. This approach reduced 4T1 breast cancer cell migration and invasion while enhancing macrophage-mediated tumor phagocytosis [144].

These investigations collectively underscore the translational potential of EV-based siRNA delivery systems in overcoming biological barriers, improving targeting specificity, and minimizing off-target effects, thereby advancing precision cancer therapy.

#### 4.3.2. mRNA Delivery

Although lipid and polymeric nanocarriers have been widely investigated for mRNA delivery, persistent challenges surrounding nonspecific biodistribution and systemic immune activation remain unresolved. Extracellular vesicles (EVs), as natural mRNA carriers, offer a promising alternative with enhanced biocompatibility and targeting precision. For instance, Liu et al. engineered an inhalable cancer vaccine (IL-12-Exo) through the exosomal encapsulation of interleukin-12 (IL-12) mRNA. In murine orthotopic lung cancer models, IL-12-Exo activated potent localized and systemic antitumor immune responses without inducing measurable toxicity. The inhalation route enabled direct pulmonary delivery, outperforming conventional systemic administration methods [145]. Xing et al. engineered EVs to encapsulate gasdermin D-N-terminal (GSDMD-N) mRNA, selectively targeting HER2+ breast cancer cells. This approach achieved the efficient translation of GSDMD-N mRNA within tumors, inducing pyroptosis and suppressing tumor growth [146]. Further optimizing mRNA packaging, Liu et al. established that ARRDC1-p53 (ARP) and CD63-p53 (CDP) fusion constructs markedly improved p53 mRNA/protein incorporation into small EVs (sEVs). Engineered sEVs from both constructs efficiently transported functional p53 to H1299 lung cancer cells, inhibiting proliferation while triggering apoptosis [147].

Mesenchymal stem cell (MSC)-derived EVs exhibit precise tissue-targeting mRNA delivery capabilities. Wu et al. successfully loaded miR-138-5p into human umbilical cord MSC-derived small EVs (hucMSC-sEVs) through electroporation, achieving efficient cellular uptake in gastric cancer models. The encapsulated microRNA suppressed ATG2A-dependent autophagic flux, markedly attenuating tumor progression and metastatic dissemination [72]. Similarly, Zhou et al. engineered bone marrow mesenchymal stem cell-derived extracellular vesicles (BMSC-EVs) for the targeted delivery of miR-766-3p to colorectal cancer cells. The miRNA cargo effectively downregulated the MYC/CDK2 oncogenic axis, concurrently inhibiting malignant proliferation, migratory capacity, and invasive potential while inducing apoptotic processes [148]. To enhance targeting specificity, Zhou et al. functionalized BMSC-EVs with integrin α5-targeting peptides for the dual payload delivery of miR-138-5p and pirfenidone (PFD) in pancreatic cancer models. The miRNA disrupted FERMT2-TGFBR1 complex assembly, thereby abrogating TGF-β signaling activation, whereas PFD concurrently remodeled cancer-associated fibroblasts (CAFs) to suppress their protumorigenic activities [149].

These studies collectively highlight EVs as versatile platforms for mRNA delivery, combining natural tropism, reduced immunogenicity, and engineering adaptability to overcome biological barriers and enhance therapeutic precision.

#### 4.3.3. CRISPR/Cas9

The CRISPR/Cas9 system, a pioneering genome-editing platform, demonstrates transformative potential in gene therapy but faces clinical limitations due to delivery inefficiency and safety concerns. Recent advances in extracellular vesicle (EV)-based delivery systems overcome these limitations through improved molecular targeting accuracy and reduced immunogenicity. For instance, Liu et al. engineered dual-modified EVs decorated with Angiopep-2 (Ang) and transactivator of transcription (TAT) peptides to co-deliver Cas9 protein and sgRNA complexes. These EVs not only traversed the blood–brain barrier (BBB) but also penetrated glioblastoma tissues, achieving an unprecedented 67.2% gene-editing efficiency in tumor cells. This platform represents a breakthrough in overcoming anatomical barriers for central nervous system (CNS) malignancies [150]. Further advancing combinatorial therapies, researchers developed PD-1- and Angiopep-2-modified EVs to co-target PLK1 and VEGF via multiplexed CRISPR editing. This approach achieved intratumoral dual-gene knockout rates of 58.6% (PLK1) and 52.7% (VEGF), inducing tumor apoptosis and anti-angiogenic effects while synergizing immune checkpoint blockade with gene editing—a paradigm shift in glioblastoma treatment [151].

Additionally, Qian et al. harnessed chimeric antigen receptor (CAR)-modified epithelial cell-derived EVs as a bioengineered CRISPR/Cas9 delivery platform. CAR-EVs selectively accumulated in tumors and efficiently released MYC-targeting CRISPR components, suppressing oncogene expression in vitro and in vivo. This strategy highlights EVs’ capacity to integrate tumor-specific homing mechanisms (via CAR) with genome-editing payloads, offering a theranostic platform with reduced off-target risks [152]. Collectively, these studies demonstrate EV-based CRISPR delivery systems as a robust, multifunctional toolkit to enhance editing precision, bypass biological barriers, and mitigate systemic toxicity, thereby accelerating translational applications in precision oncology.

### 4.4. Application in Immunotherapy

Immunotherapy has revolutionized oncology, yet suboptimal therapeutic response rates and immune-mediated adverse events persist as critical limiting factors in clinical translation. Emerging evidence suggests that extracellular vesicles (EVs), through their capacity to remodel the immunosuppressive tumor microenvironment (TME), offer novel strategies to enhance immunotherapeutic efficacy [153]. For instance, Peng et al. designed tumor cell-derived EVs that co-delivered STAT3-silencing siRNA and doxorubicin (siSTAT3-DOX@TEV). These EVs selectively accumulated in tumors, downregulating STAT3 expression, inducing immunogenic cell death, and reprogramming the TME by increasing M1 macrophages, CD4^+^, and CD8^+^ T-cell infiltration, thereby synergizing chemotherapy with immune activation [154]. Similarly, Li et al. developed multifunctional EVs encapsulating manganese-based superoxide dismutase (SOD), the STING agonist diABZI-2, and siSTAT3 for malignant pleural effusion (MPE) treatment. Systemic administration eradicated MPE and pleural tumors in aggressive mouse models by reversing immunosuppression, triggering systemic antitumor immunity, and establishing durable immune memory, with 100% survival rates. Notably, this approach also demonstrated immunomodulatory effects in patient-derived MPE samples, underscoring its translational potential [155].

Targeting the cGAS-STING pathway, a pivotal innate immune sensor of cytosolic DNA, represents a promising avenue for immunotherapy [156]. Ma et al. developed an EV-mediated protein delivery platform (IDEA) for the targeted transport of cyclic GMP-AMP synthase (cGAS). cGAS-EVs activated interferon signaling across multiple syngeneic tumor models, enhancing antitumor immunity and alleviating TME immunosuppression. The co-administration of cGAS-loaded EVs with immune checkpoint inhibitors synergistically enhanced antitumor responses, highlighting the synergy between innate immune activation and adaptive immune potentiation [157]. Qian et al. similarly leveraged mesenchymal stem cell (MSC)-derived EVs to deliver STING agonists, inducing IFNβ expression in monocytes and augmenting antitumor responses in vivo [158]. To achieve spatiotemporal control of EV activity, Zhang et al. designed a “nanocloak” by encapsulating bacteria-derived EVs (BEVs) within a manganese dioxide (MnO_2_) shell (cBEV). This interface protected BEVs from degradation while enabling Mn^2+^-triggered cGAS-STING activation and dendritic cell maturation upon MnO_2_ dissolution, thereby enhancing antigen-specific T-cell responses [159].

Innovative combinatorial strategies further expand EV utility. Wei et al. engineered exosomes co-loaded with manganese-doped iron oxide nanoparticles (MnIO), the ferroptosis inhibitor GW4869, and buthionine sulfoximine (BSO). This system disrupted iron and redox homeostasis, achieving 29.57% ID/g tumor iron retention and promoting ferroptosis, while the GW4869 inhibition of iron efflux and BSO-induced glutathione depletion synergized to amplify immunogenic tumor cell death [160]. Jayasinghe et al. engineered EV surfaces through inverse electron-demand Diels–Alder (iEDDA) conjugation chemistry to present immunomodulatory ligands. These modified EVs reprogrammed the tumor microenvironment (TME) into an antitumorigenic phenotype, demonstrating enhanced efficacy in reducing the tumor burden and prolonging survival in metastatic models compared to unencapsulated ligands [161]. Gao et al. developed prostate-specific membrane antigen (PSMA)-targeting EVs expressing scFv fragments and loaded with gasdermin D-N-terminal (GSDMD), which induced pyroptosis in PSMA^+^ prostate cancer cells, activating antitumor immunity and suppressing tumor growth [162].

Garaeva et al. elucidated that grapefruit-derived EVs (GEVs) loaded with HSP70 sensitized colorectal cancer cells to cytotoxic lymphocytes and NK cells, requiring 20-fold lower HSP70 doses than free protein. GEVs-HSP70 prolonged survival, reduced tumor volume, and decreased immunosuppressive cytokines (TGF-β1, IL-10) in murine models, suggesting a potent yet low-toxicity immunotherapeutic approach [163]. Guo et al. engineered M1 macrophage-derived EVs (REV@SR780Fe@LEV-RS17) co-loaded with RS17-targeting peptides, SR780Fe nanoparticles, and reveromycin A (REV). In the acidic TME, SR780Fe degraded to release SR780 (a photosensitizer) and Fe^3+^, inducing photodynamic therapy and ferroptosis via Fenton reactions. Concurrently, REV activated the cGAS-STING pathway, repolarizing tumor-associated macrophages (TAMs), expediting dendritic cell immunogenic maturation, and enhancing cytotoxic T-lymphocyte infiltration, thereby achieving tri-modal immuno-therapeutic synergy [164].

These advances underscore EVs as versatile platforms for multimodal immunotherapy, capable of targeting immune checkpoints, activating innate and adaptive immunity, and integrating with conventional therapies to overcome resistance and improve clinical outcomes.

### 4.5. Application in Photothermal Therapy

Photothermal therapy (PTT), which employs light-responsive agents to generate localized heat for tumor ablation, has emerged as an emerging anticancer strategy. However, effective PTT requires the precise delivery of photothermal agents to tumor tissues while minimizing off-target effects. Extracellular vesicles (EVs), with their inherent biocompatibility and tumor-targeting capabilities, have recently been harnessed as advanced carriers for photothermal nanomaterials. For instance, Sato et al. engineered a cationic polyethylene glycol (PEG)-modified gold nanostar (GNS)-EV hybrid system (EV-GNS). Compared to PEG-GNS alone, EV-GNS exhibited superior photothermal conversion efficiency and enhanced cellular uptake. Upon near-infrared (NIR) laser irradiation, EV-GNS induced rapid hyperthermia and laser-responsive cytotoxicity, demonstrating the potential of EV-mediated photothermal delivery [165]. Similarly, Bi et al. developed an EV-based platform (EPM) co-loaded with melanin (a natural photothermal agent) and paclitaxel–albumin (PA) using 4T1 breast cancer-derived EVs. EPM was efficiently internalized by both the tumor cells and dendritic cells, enabling dual-modality photothermal and chemotherapy. Under NIR irradiation, EPM generated robust photoacoustic signals and localized hyperthermia, synergizing with PA to enhance CD8^+^ T-cell infiltration and antitumor immunity in murine models [166].

To further integrate PTT with immunomodulation, Lu et al. encapsulated siRNA targeting PAK4 (a pro-survival kinase) and photoactivatable polyethyleneimine (PEI) within M1 macrophage-derived EVs. This system achieved efficient PAK4 silencing and triggered immunogenic phototherapy upon light exposure, effectively suppressing tumor growth while maintaining low systemic toxicity and high biocompatibility [167]. Lee et al. advanced tumor-specific delivery by developing melanoma-derived exosomes loaded with indocyanine green (ICG) and camptothecin (CPT), encapsulated in a perfluorocarbon (PFC) nanoplatform (ICFES). In syngeneic melanoma models, ICFES exhibited prolonged intratumoral retention and targeted accumulation compared to free ICG. Following NIR irradiation, ICFES-mediated photochemical therapy suppressed tumor growth by 15-fold relative to free CPT, achieving potent efficacy without systemic toxicity [168].

These studies collectively highlight EVs as versatile platforms for enhancing photothermal therapy through improved tumor targeting, combinatorial therapeutic payloads, and the synergistic activation of antitumor immunity, thereby addressing critical limitations of conventional PTT approaches.

### 4.6. Application in Cancer Vaccines

Cancer vaccines, designed to elicit systemic tumor-specific immune responses, represent a promising frontier in immunotherapy. Tumor-derived extracellular vesicles (TEVs), naturally enriched with tumor-associated antigens, have emerged as compelling platforms for vaccine development due to their intrinsic immunogenicity and biocompatibility. For example, Dang et al. engineered red blood cell-derived EVs (RBCEVs) functionalized with DEC-205-targeting ligands to deliver ovalbumin (OVA) antigens to splenic dendritic cells (DCs). This approach robustly enhanced DC-mediated antigen processing and T-cell activation, eliciting potent antitumor immunity in murine models [169]. Wang et al. pioneered a bioengineered vaccine platform through the covalent conjugation of Panax ginseng-derived extracellular vesicle-like particles (G-EVLPs) with patient-specific tumor-associated antigens. The resulting hybrid nanoparticles (HM-NPs) promoted the DC phagocytosis of tumor antigens via TLR4 signaling, driving DC maturation and establishing durable tumor protection, thereby highlighting the potential of plant-derived nanomaterials in personalized immunotherapy [170].

To mitigate the intrinsic immunosuppressive properties of TEVs, Han et al. attenuated TEVs by inhibiting YAP signaling and autophagy while enriching tumor antigens and adjuvants. The resulting “attenuated immunogenic TEVs” (AI-TEVs) elicited robust tumor-specific and long-lasting immunity, functioning effectively as both prophylactic and therapeutic vaccines against recurrent cancers [171]. Similarly, researchers masked TEV surfaces with iron oxyhydroxide (FeOOH) nanocomposites to block CD47-mediated “don’t eat me” signals. These cloaked TEVs (mTEVs) were more efficiently phagocytosed by DCs and macrophages, releasing tumor antigens within lysosomes to activate antitumor immunity. This strategy enhanced tumor vaccination efficacy in preclinical models and clinical samples, underscoring its translational relevance [172].

To optimize lymph node and tumor dual targeting, Wang et al. generated TEVs from vesicular stomatitis virus (VSV)-infected 4T1 tumor cells. These viral antigen-presenting EVs (vEVs) accumulated in tumors and lymph nodes, triggering innate and adaptive immune responses against both tumor and viral antigens, thereby suppressing tumor progression [173]. By using a similar approach, the same group developed cGAMP-loaded vEVs (cGAMP@vEVs), which synergistically activated the STING pathway in lymph nodes and reprogrammed the tumor microenvironment (TME). cGAMP@vEVs promoted tumor-specific CD8^+^ T-cell expansion, enhanced cytotoxic T-lymphocyte (CTL) infiltration, and established a self-sustaining antitumor immune cycle, effectively inhibiting tumor growth, metastasis, and recurrence [174].

Recent studies also demonstrate TEVs’ potential to enhance viral vector-based vaccines. As a representative example, EVs engineered to encapsulate adeno-associated virus (AAV) vectors expressing tumor-associated antigens (e.g., ovalbumin [OVA], tyrosinase-related protein 1 [TRP-1]) demonstrated superior antigen-specific CD8^+^ T-cell priming efficacy compared to conventional AAV delivery platforms. Crucially, EVs-AAV formulations achieved equivalent therapeutic outcomes at 2-log lower dosing regimens, substantially reducing vector-induced hepatotoxicity as quantified by serum ALT/AST levels [175]. Similarly, Mathlouthi et al. utilized melanoma-derived EVs to deliver oncolytic viruses, which improved tumor-targeted viral delivery, amplified intratumoral viral loads, and enhanced immunogenic cell death. This EV-mediated strategy not only increased T-cell infiltration but also shielded viruses from host immune clearance, yielding superior antitumor effects over standalone viral therapies [176].

Collectively, these innovations position TEVs as versatile, multifunctional vaccine platforms capable of delivering tumor antigens, modulating immune checkpoints, and synergizing with viral vectors to overcome traditional limitations in cancer vaccination (Table 2).

## 5. Clinical Advancements in Extracellular Vesicle-Based Drug Delivery Systems

While extracellular vesicles (EVs) have been extensively investigated for diagnostic and prognostic applications in clinical trials, their therapeutic potential as drug carriers in oncology remains underexplored, with limited interventional studies reported to date [179].

### 5.1. Completed Clinical Trials

Pioneering studies focused on dendritic cell-derived exosomes (Dex) demonstrated early translational promise. In a Phase I trial involving patients with terminal cancer, first-generation Dex (without IFN-γ priming) enhanced natural killer (NK) cell-mediated antitumor activity, establishing preliminary safety and immunogenicity. A subsequent Phase II trial evaluated IFN-γ-primed Dex loaded with MHC class I/II-restricted tumor antigens in patients with unresectable non-small-cell lung cancer (NSCLC) receiving post-induction chemotherapy. The treatment correlated with prolonged disease stabilization, yielding a significantly improved 4-month progression-free survival (PFS) rate. This study demonstrated the ability of Dex to enhance NK cell activity and potentiate adaptive antitumor immune responses in treatment-refractory advanced NSCLC [180]. Another Phase I trial investigating ascites-derived exosomes (Aex) co-administered with granulocyte-macrophage colony-stimulating factor (GM-CSF) enrolled 40 patients with chemotherapy-refractory metastatic colorectal adenocarcinoma, randomized 1:1 to Aex monotherapy or combination therapy. Both arms demonstrated favorable safety profiles, with the AEx/GM-CSF cohort inducing polyfunctional CD8+ T-cell responses, providing proof of concept for this ascites-based immunotherapeutic platform in refractory disease [181]. Additionally, autologous Dex pulsed with MAGE-3 peptides was tested in patients with stage III/IV melanoma. The Phase I trial confirmed the absence of grade ≥2 toxicities and the feasibility of large-scale exosome production, underscoring the clinical scalability of EV-based vaccines [182].

In recent years, drug-loaded vesicle-based therapies have demonstrated promising outcomes in clinical trials targeting malignant pleural effusion (MPE) and cholangiocarcinoma. Guo et al. utilized methotrexate-loaded tumor cell-derived microparticles (MTX–TMPs) for the treatment of MPE. In a clinical study involving 11 patients with advanced lung cancer-associated MPE, MTX–TMPs achieved an objective response rate of 90.91% and a median survival of 240 days, confirming the feasibility and safety of TMPs-MTX preparation and administration. Notably, no grade 3 or higher toxicities were observed [183]. Similarly, Gao et al. administered MTX–TMPs via intrabiliary infusion in patients with cholangiocarcinoma. Mechanistically, MTX–TMPs recruited neutrophils to tumor sites through the UDP-glucose (UDPG) and complement component 5 (C5) pathways, thereby eliciting antitumor immunity. Furthermore, MTX–TMPs triggered cholangiocarcinoma cell pyroptosis via the gasdermin E (GSDME)-dependent pathway. Among 20 patients with advanced hilar cholangiocarcinoma, approximately 30% exhibited radiologic improvement in biliary obstruction, while 50% showed alleviated jaundice and improved liver function after the first treatment cycle. Safety assessments revealed no significant changes in blood counts or hepatic or renal function, and no adverse events such as abdominal pain, nausea, or vomiting were reported, underscoring the favorable safety profile of this approach [184].

### 5.2. Ongoing Clinical Trials

Current investigations aim to expand EV therapeutic applications. A Phase I clinical trial (NCT03608631) is investigating mesenchymal stem cell-derived exosomes (iExosomes) engineered to deliver KrasG12D siRNA in patients with metastatic pancreatic ductal adenocarcinoma (PDAC) harboring KrasG12D mutations. This study aims to establish the maximum tolerated dose (MTD) and characterize dose-limiting toxicities (DLTs), with a concurrent evaluation of KRAS oncogene-silencing therapeutic efficacy [185]. Another trial (NCT01294072) is exploring plant-derived EVs for targeted curcumin delivery in colorectal cancer. The study aims to characterize curcumin’s immunomodulatory and metabolic effects on malignant and normal colonic tissues, including cytokine profiling, immune cell dynamics, and phospholipid metabolism in surgical specimens. Subgroup analyses will further investigate glucose metabolism alterations via 13C-glucose tracing in patients receiving EV-curcumin preoperatively [186] (Table 3).

These trials collectively highlight the evolving landscape of EV-based therapeutics, bridging preclinical innovation with clinical validation to address unmet needs in precision oncology.

### 5.3. Current Limitations in Clinical Research Progress

Extracellular vesicles (EVs), an emerging class of drug delivery vehicles, hold immense promise for cancer therapy. However, their clinical translation faces multifaceted challenges spanning technical, ethical, and regulatory domains.

From a technical perspective, critical hurdles include the need to optimize EV preparation and purification methods to ensure consistent quality and functionality, as current protocols risk altering their structural integrity and biological activity. Stability during storage and in physiological environments remains another major obstacle, as EVs are prone to enzymatic degradation and oxidation. Most critically, safety concerns necessitate long-term clinical monitoring to evaluate potential side effects, particularly given the risk that EVs may carry tumor-associated molecules, inadvertently promoting tumorigenesis or metastasis [189].

Ethically and legally, safeguarding participant privacy and ensuring informed consent in EV-related studies are non-negotiable priorities. Additionally, the involvement of multiple stakeholders—researchers, patients, and commercial entities—demands transparent frameworks to equitably distribute benefits and protect rights.

Regulatory challenges further complicate clinical adoption. Standardized clinical trial protocols are urgently needed to validate the efficacy and safety of EV-based therapies, which remain in early developmental stages. Equally important is establishing rigorous quality control systems to standardize source material selection, production processes, and final product characterization, ensuring reproducibility and safety across batches [190].

In summary, unresolved technical limitations, ethical ambiguities, or regulatory gaps could render EVs unsuitable for clinical research. Future efforts must integrate multidisciplinary strategies to address these barriers, accelerating the translation of EV-based therapies into mainstream clinical practice.

## 6. Conclusions and Perspectives

This review underscores the translational potential of extracellular vesicles (EVs) as drug delivery systems in oncology, highlighting the recent advances in EV isolation techniques, cargo-loading strategies, and their preclinical-to-clinical translation. The optimization of preparation methods not only enhances EV yield but also tailors their physicochemical properties to improve drug delivery efficiency. Concurrently, innovative loading approaches—including genetic engineering, chemical modification, and physical encapsulation—significantly increase drug payloads while preserving EV structural integrity and biofunctionality. These methodological refinements are pivotal for maximizing therapeutic efficacy and minimizing off-target effects.

EVs from distinct biological sources demonstrate unique therapeutic advantages in oncology. For example, immune cell-derived EVs exhibit intrinsic antitumor activity, tumor cell-derived EVs inherently target malignant tissues, and plant- or milk-derived EVs display exceptional oral bioavailability. These extraordinary properties substantially broaden the applicability of EVs as versatile drug delivery platforms. Furthermore, EVs originating from diverse sources hold immense potential for personalized medicine and combined therapeutic strategies. By leveraging artificial intelligence (AI) and big data, clinicians could identify optimal EV sources and drug-loading regimens tailored to individual patients, thereby advancing precision oncology [191].

The large-scale clinical translation of EVs still faces multiple challenges that must be addressed. Key hurdles include establishing GMP-compliant large-scale production methods, optimizing storage protocols to preserve EV integrity and functionality, determining optimal administration routes, and ensuring the long-term safety of EV-based therapies in clinical settings [192]. To overcome these barriers, the development of standardized quality control systems and rigorous regulatory oversight for clinical trial design and approval processes will be critical to enabling the widespread application of EV therapeutics.

In summary, EVs, as natural drug delivery systems, exhibit remarkable advantages in cancer therapy but are accompanied by critical challenges. Future advancements could focus on three key strategies: (1) developing simplified and efficient EV isolation techniques, (2) inventing optimized drug-loading methodologies to enhance cargo capacity, and (3) selecting EV sources with inherent therapeutic potential while integrating advanced materials for functional modifications. Through these approaches, EVs are poised to emerge as pivotal tools in next-generation cancer therapeutics, offering novel solutions to improve treatment efficacy, minimize off-target effects, and advance personalized medicine.

## Figures and Tables

**Figure 1 ijms-26-04835-f001:**
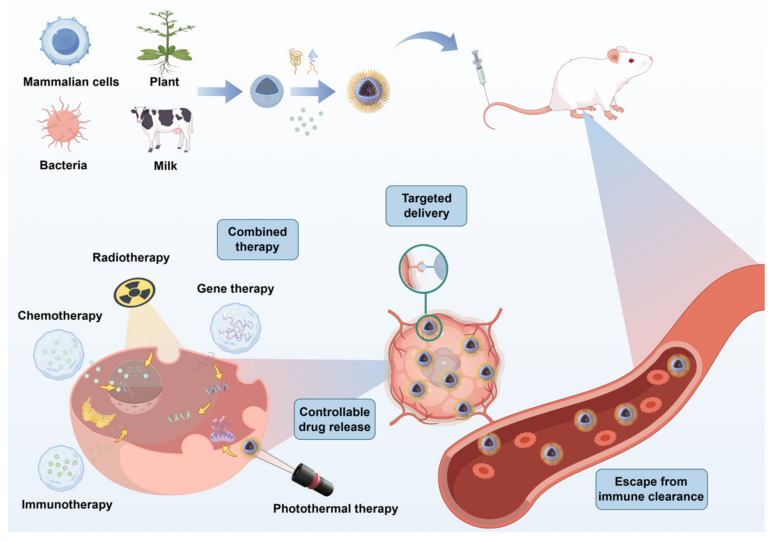
Extracellular vesicles (EVs) have emerged as transformative nanocarriers in oncology, leveraging inherent biological advantages, including prolonged systemic circulation, intrinsic tumor-homing properties, multi-level biodistribution capacities, and stimulus-responsive release mechanisms. These characteristics synergistically enhance therapeutic precision through tumor-specific accumulation while reducing off-target effects via spatiotemporal payload control, establishing EVs as engineered platforms reconciling targeted delivery with endogenous biocompatibility.

**Figure 2 ijms-26-04835-f002:**
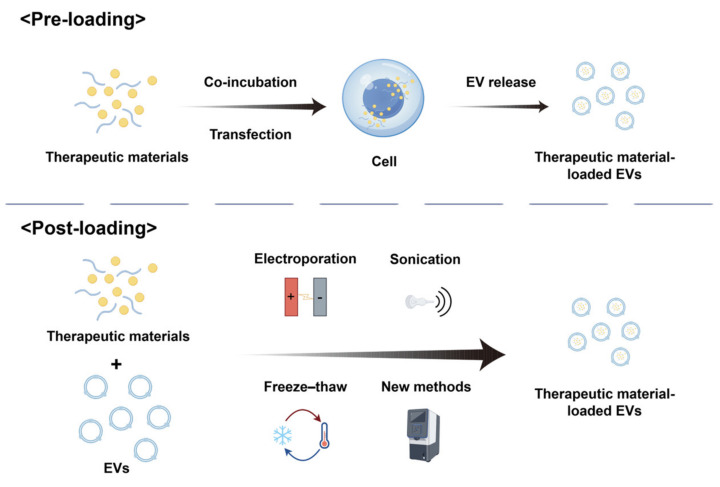
Drug-loading strategies for extracellular vesicles (EVs): pre-loading and post-loading approaches.

**Figure 3 ijms-26-04835-f003:**
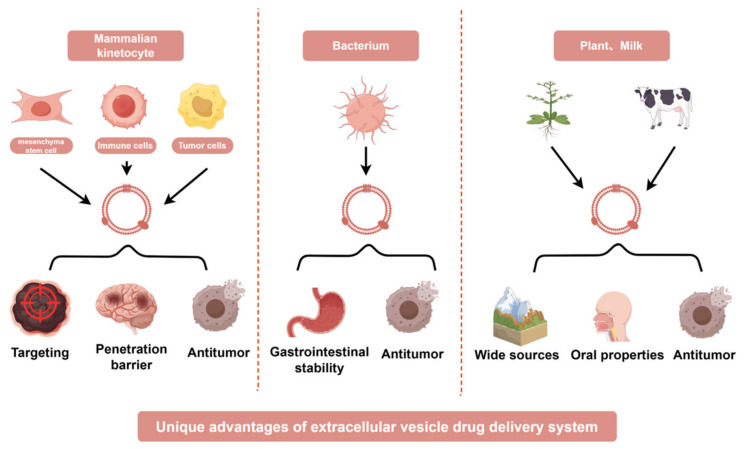
Extracellular vesicles derived from diverse biological sources exhibit unique advantages as drug delivery systems in cancer therapy. EV sources include the following: Mammalian cells: mesenchymal stem cells (MSCs), immune cells (e.g., dendritic cells, T cells), and cancer cells (e.g., tumor-derived EVs); Bacteria: engineered bacterial outer membrane vesicles (OMVs); Plants: plant-derived exosome-like nanoparticles (PDENs); Milk: bovine milk-derived EVs. The distinct benefits of these EVs arise from their intrinsic biological properties. For example, mammalian cell-derived EVs inherit natural targeting capabilities and biocompatibility, bacterial EVs can be engineered to carry immunostimulatory molecules, plant-derived EVs offer low immunogenicity and cost-effective scalability, and milk-based EVs provide a biocompatible platform for large-scale production. This diversity enables tailored therapeutic strategies to enhance drug delivery precision, minimize off-target effects, and address tumor heterogeneity.

**Figure 4 ijms-26-04835-f004:**
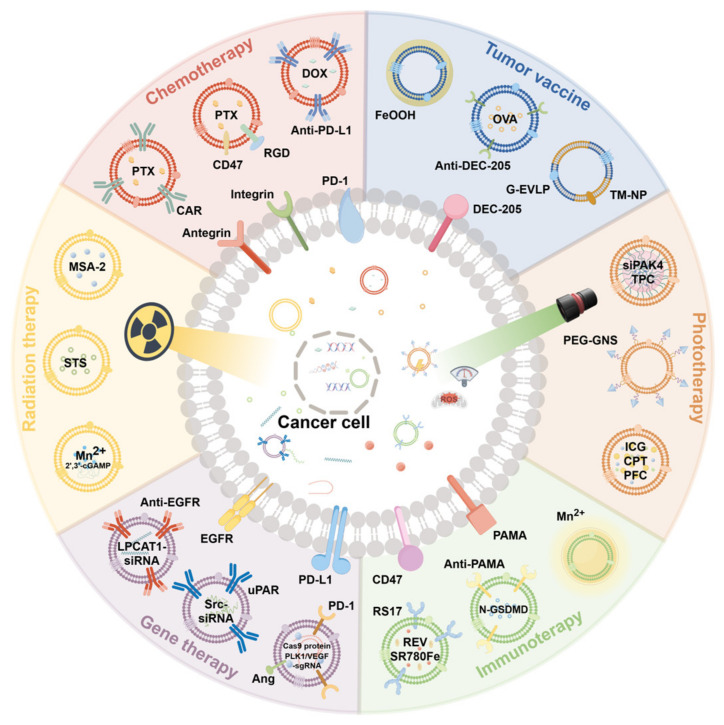
Extracellular vesicles (EVs) serve as versatile drug delivery systems for chemotherapy, radiotherapy, gene therapy, immunotherapy, photothermal therapy, and cancer vaccines.

**Table 1 ijms-26-04835-t001:** The advantages of EVs from diverse sources as drug delivery systems in cancer therapy.

EV Source	EV Type	Key Advantages/Mechanisms	Cancer Model	Ref.
Rat adipose-derived MSCs	EVs	Specific targeting and penetration into bladder cancer tissues for RNA agent delivery	Mouse bladder cancer model	[68]
Mouse bone marrow MSCs	EVs	Osteosarcoma targeting via SDF1-CXCR4 axis; reduced chemotherapy-induced toxicity	Mouse osteosarcoma model	[69]
Human placental MSCs	EVs	Anti-angiogenic effects; inhibition of tumor cell proliferation and migration	Mouse breast cancer model	[71]
Human NK cells	EVs	Induced tumor apoptosis via serine/threonine kinase dephosphorylation; active liver cancer targeting	Mouse hepatocellular carcinoma model	[77]
CD8^+^ T cells	EVs	Delivered granzyme B, perforin, and IFNγ to induce tumor apoptosis	Breast cancer cells	[76]
Human memory-like NK cells	EVs	Triggered caspase-dependent apoptosis	Mouse NSCLC and gastric cancer models	[72]
Rat adipose-derived MSCs	EVs	Specific targeting and penetration into bladder cancer tissues for RNA agent delivery	Mouse bladder cancer model	[68]
Mouse/human eosinophils	EVs	Arrested tumor cell cycle at G0/G1 phase; suppressed proliferation and tumor spheroid formation	Mouse melanoma model	[80]
Human colorectal cancer cells	EVs	Tumor-specific delivery of miRNA therapeutics	Mouse colorectal cancer model	[81]
Mouse breast cancer cells	EVs	Intrinsic breast cancer tissue targeting	Mouse breast cancer model	[83]
Canine glioma cells	EVs	Penetrated blood–brain barrier (BBB); selective accumulation in brain tumors	Mouse glioma model	[85]
Melanoma cells	EVs	Targeted primary melanoma and lung metastatic nodules	Mouse melanoma model	[84]
Lactobacillus reuteri	EVs	Modulated apoptotic signaling; oral administration with gastrointestinal stability	Mouse NSCLC model	[89]
Lactobacillus paracasei	EVs	Inhibited HIF-1α-mediated glycolysis	Mouse colon cancer model	[90]
Lactobacillus plantarum	EVs	Impaired p53 desuccinylation, suppressing glycolysis and proliferation	Mouse colon cancer model	[91]
Bifidobacterium longum	EVs	Enhanced p53 Ser15 phosphorylation; sensitized ovarian cancer cells to carboplatin	Ovarian cancer cell model	[92]
Escherichia coli	OMVs	Activated CD8^+^ T-cell infiltration and antitumor immunity	Mouse bladder/breast cancer models	[93]
Photosynthetic bacteria	OMVs	Polarized macrophages to M1 phenotype; promoted pro-inflammatory cytokine release	Mouse breast cancer model	[94]
Catharanthus roseus (plant)	PELNVs	Mitigated chemotherapy-induced immunosuppression by modulating TNF-α/NF-κB/Spi-B signaling	In vitro	[101]
Perilla leaves	EVs	Selectively inhibited breast cancer cell proliferation and invasion	Human breast cancer cell model	[96]
Celery	EVs	Blocked PD-L1/PD-1 interaction; reversed T-cell suppression	Mouse lung cancer model	[97]
Platycodon root	EVs	Induced ROS-mediated apoptosis; repolarized TAMs to M1 phenotype; tumor targeting	Mouse breast cancer model	[98]
Lemon	EVs	Inhibited PI3K/AKT and MAPK/ERK pathways; suppressed proliferation and metastasis	Human breast cancer cell model	[103]
Ginger	EVs	Delivered cytotoxic gingerols/shogaols; induced cell cycle arrest and p53-mediated apoptosis	Mouse melanoma model	[104]
Basil plant	EVs	Efficient tumor uptake; promoted apoptosis	Human pancreatic cancer cells	[105]
Garlic	EVs	Activated caspase-dependent apoptosis in multiple cancer types	Human liver, neuroblastoma, pancreatic, glioblastoma, prostate cancers; HUVECs	[99]
Cow milk	EVs	Protected polyphenols from degradation; enhanced therapeutic efficacy	Human breast cancer cells	[108]

**Table 2 ijms-26-04835-t002:** Application of extracellular vesicles as drug delivery systems in cancer therapy.

Method	Drug	EV Source	EV Type	Therapeutic Effects	Cancer Type	Ref.
Chemotherapy	DOX	Bitter melon	EVs	Suppressed tumor growth and alleviated cardiotoxicity	Breast cancer	[114]
	DOX	Dendritic cells	EVs	Reduced tumor angiogenesis and induced apoptosis	Glioma	[115]
	DOX	FreeStyle 293F cells	PD-L1 antibody-engineered Fc-EVs	Enhanced tumor suppression with reduced systemic toxicity	Melanoma	[116]
	DOX	Rhodiola rosea	pYEEIE-modified exosome-like vesicles	Tumor-targeted delivery and growth inhibition	Melanoma	[117]
	DOX	Human breast cancer cells	EVs	Synergistic tumor suppression and toxicity mitigation	Breast cancer	[118]
	PTX	T cells	CAR-modified EVs	Targeted tumor inhibition and reduced adverse effects	Lung cancer	[119]
	PTX	HEK293 cells	RGD- and CD47-engineered EVs	Tumor-specific targeting and growth suppression	Pancreatic ductal adenocarcinoma	[120]
	PTX	Cow milk	Transferrin-conjugated EVs	Enhanced drug delivery and tumor suppression	Liver cancer	[121]
	PTX	Lung cancer cells	Large EVs	Inhibited autophagy pathways and induced apoptosis	Lung cancer	[122]
	PTX + SUR-siRNA	HEK293T cells	CD44-modified EVs	Dual suppression of tumor growth and chemoresistance	Breast cancer	[123]
	PTX	Human MSCs	EVs	Tumor growth inhibition	Breast cancer	[124]
	Carboplatin	NK cells	EVs	Enhanced antitumor efficacy	Lung cancer	[126]
	Oxaliplatin	Human colorectal cancer cells	Porphyrin-modified EVs	Tumor suppression with reduced toxicity	Colorectal cancer	[127]
	Gemcitabine	Human MSCs	EVs	Promoted apoptosis and reduced off-target effects	Pancreatic cancer	[128]
	Gemcitabine	Human pancreatic cancer cells	Exosome–liposome hybrid nanoparticles	Precision targeting and tumor growth inhibition	Pancreatic ductal adenocarcinoma	[129]
	MTX	hAMSCs	CD19-modified exosomes	Blood–brain barrier (BBB) penetration and tumor-specific delivery	CNS lymphoma (CNSL)	[131]
	Vinorelbine	Human umbilical endothelial cells	GE11 peptide-modified EVs	Tumor-targeted delivery with minimal toxicity	Lung cancer	[130]
	5-FU	Broccoli	EVs	Enhanced chemosensitivity and tumor suppression	Colorectal cancer	[132]
Radiotherapy	MSA-2	Human breast cancer cells	EVs	Reduced radioresistance and optimized FLASH radiotherapy	Breast cancer	[133]
	STS	Human esophageal cancer cells	EVs	Improved radiosensitivity and therapeutic efficacy	Esophageal cancer	[134]
	Mn2+,2′,3′-cGAMP	Melanoma cells	EVs	Overcame immunosuppression and enhanced radioimmunotherapy	Melanoma	[135]
Gene Therapy	siSTAT3	Kiwifruit	EVs	Tumor growth inhibition	Non-small cell lung cancer (NSCLC)	[177]
	DDHD1-siRNA	Orange juice	EVs	Suppressed tumor proliferation	Colorectal cancer	[138]
	Mstn-siRNA	Red blood cells	EVs	Reduced cachexia-associated toxicity	Cancer cachexia	[177]
	TRAIL	Neural stem cells	EVs	Selective tumor targeting and growth suppression	Brain cancer	[178]
	siSIRPα + siSTAT6	M1 macrophages	EVs	Dual inhibition of tumor growth and immunosuppression	Breast cancer	[144]
	siRNA (KRASG12C)	Cow milk	EVs	KRAS oncogene silencing and tumor suppression	NSCLC	[143]
	LPCAT1-siRNA	HEK293T cells	EGFR scFv-modified EVs	BBB penetration and tumor-specific delivery with low toxicity	Lung cancer brain metastasis	[138]
	siR-DG	Human breast cancer cells	Chiral graphene quantum dot/pH-responsive peptide-modified EVs	Targeted tumor cell apoptosis	Breast cancer	[137]
	miR-34a + CDAmiR	Human MSCs	EGFRvIII antibody-modified EVs	Precision targeting and apoptosis induction	Glioblastoma	[141]
	Src siRNA	Human MSCs	uPA peptide-modified EVs	Tumor cell apoptosis and metastasis inhibition	Breast cancer	[142]
	IL-12 mRNA	—	EVs	Enhanced antitumor immunity with reduced toxicity	Lung cancer	[139]
	GSDMD-N mRNA	—	EVs	Induced pyroptosis and tumor suppression	Breast cancer	[146]
	p53 mRNA	HEK293T cells	ARRDC1- and CD63-engineered EVs	Restored p53 function and tumor apoptosis	Lung cancer	[147]
	miR-13896	Human MSCs	EVs	Autophagy inhibition and tumor growth suppression	Gastric cancer	[72]
	miR-766-3p	Human MSCs	EVs	Suppressed proliferation and promoted apoptosis	Colorectal cancer	[148]
	Cas9/sgRNA	HEK293T cells	Angiopep-2- and TAT peptide-modified EVs	BBB penetration and tumor growth inhibition	Glioblastoma	[147]
	Cas9/sgRNA	HEK293T cells	PD-1- and angiopep-2-modified EVs	Anti-angiogenic effects and tumor apoptosis	Glioblastoma	[151]
	Cas9/sgRNA	HEK293T cells	CAR-modified EVs	Reduced proliferation and enhanced apoptosis in B-cell malignancies	B-cell malignancies	[152]
Immunotherapy	siSTAT3 + DOX	Mouse breast cancer cells	EVs	Reversed immunosuppressive TME and suppressed tumor growth	Breast cancer	[154]
	SOD + diABZI-2 + siSTAT3	M1 macrophages	EVs	Alleviated immunosuppressive TME and inhibited tumor progression	Malignant pleural effusion	[155]
	cGAS	HEK293T cells	EVs	Activated cGAS-STING pathway to remodel TME and suppress tumorigenesis	Colon cancer	[157]
	Activated STING	MSCs	EVs	Induced IFNβ expression and enhanced antitumor immunity	Breast cancer	[158]
	—	Bacteroides fragilis	Responsive nanocloak-coated EVs	Stimulated cGAS-STING signaling and inhibited tumor growth	Breast cancer	[159]
	MnIO + GW4869 + BSO	Mouse breast cancer cells	EVs	Promoted tumor immunogenicity and ferroptosis	Breast cancer	[160]
	N-GSDMD	HEK293T	PAMAscFv-modified EVs	Induced pyroptosis and suppressed tumor proliferation	Prostate cancer	[162]
Photothermal Therapy	HSP70	Grapefruit	EVs	Elicited antitumor immunity and suppressed tumor growth	Colon cancer	[163]
	REV + SR780Fe	M1 macrophages	RS17 peptide-engineered EVs	Enhanced T-lymphocyte infiltration and tumor suppression	Breast cancer	[164]
	—	HEK293T cells	PEG-GNS-modified EVs	Generated cytotoxic effects under laser irradiation	Melanoma	[165]
	Melanin + paclitaxel albumin (PA)	Mouse breast cancer cells	EVs	Near-infrared (NIR)-activated CD8^+^ T-cell infiltration and antitumor efficacy	Breast cancer	[166]
Cancer Vaccines	—	M1 macrophages	siPAK4- and TPC-modified EVs	Induced PAK4 silencing and immunogenic phototherapy	Melanoma	[167]
	ICG + CPT + PFC	Melanoma cells	EVs	NIR-triggered tumor suppression without systemic toxicity	Melanoma	[168]
	OVA	Mouse red blood cells	αDECab-engineered EVs	Enhanced DC phagocytosis of tumor antigens via TLR4-mediated maturation, enabling durable tumor protection	—	[169]
	Tumor antigens + G-EVLP	Ginseng	EV-like hybrid membrane particles	Boosted adaptive immunity to inhibit tumor recurrence and metastasis	Melanoma, breast cancer	[170]
	—	Mouse thymoma cells	EVs	Induced tumor-specific, long-lasting immunity against recurrence	Thymoma	[171]
	—	Mouse breast cancer cells	FeOOH-modified EVs	Activated DC and macrophage responses, prolonging survival and delaying tumor growth	Breast cancer	[172]
	—	VSV-infected mouse breast cancer cells	EVs	Accumulated in tumors and lymph nodes, triggering innate/adaptive immune responses	Breast cancer	[173]
	OVA, TRP-1	Melanoma cells	EVs	Activated antigen-specific immunity and improved survival	Melanoma	[175]
	Oncolytic virus	Melanoma cells	EVs	Enhanced lymphocyte infiltration and antitumor efficacy	Melanoma	[176]

**Table 3 ijms-26-04835-t003:** Extracellular vesicles as drug delivery systems: clinical trials in cancer.

NCT Number	EV Type	Disease Type	Intervention/ Treatment	Sponsor	Status	Ref.
NCT01159288	Dendritic cell-derived exosomes loaded with antigen	Non-Small-Cell Lung Cancer	Biological: Dex2	Gustave Roussy, Cancer Campus, Grand Paris	Completed; Phase 2	[187]
NCT03608631	Mesenchymal–stromal cell-derived exosomes loaded with siRNA against KrasG12D	Metastatic Pancreatic Adenocarcinoma; Pancreatic Ductal Adenocarcinoma	Drug: Mesenchymal–stromal cell-derived exosomes with KRAS G12D siRNA	M.D. Anderson Cancer Center	Active not recruiting; Phase 1	[185]
NCT01294072	Plant exosomes loaded with curcumin	Colon Cancer	Dietary supplement: curcumin Dietary supplement: curcumin conjugated with plant exosomes Other: no intervention	University of Louisville	Recruiting; NA	[186]
NCT05375604	Cell-derived exosomes loaded with a synthetic lipid-tagged oligonucleotide	Advanced Hepatocellular Carcinoma (HCC) Gastric Cancer Metastatic to Liver Colorectal Cancer Metastatic to Liver	Drug: CDK-004	Codiak BioSciences	Terminated; Phase 1	[188]

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
