# Peer review of "Extracellular Vesicle-Based Drug Delivery Systems in Cancer Therapy"

_ijms, 2025, doi:10.3390/ijms26104835_

Round 1
Reviewer 1 Report
Comments and Suggestions for Authors
The manuscript by Wu et al. provides an insightful and timely overview of recent advances in the application of extracellular vesicles (EVs) as carriers for therapeutic agents in cancer treatment. The authors thoroughly discuss current methods for the isolation and characterization of EVs, as well as their biological sources, offering a clear and detailed account of the state of the art in this area. A significant strength of the paper is its focus on the use of EVs in advanced therapeutic strategies, including gene therapy and combination treatments. Moreover, the authors go beyond preclinical studies by reviewing and critically evaluating clinical trials conducted to date, which adds further depth and relevance to the work. In summary, the manuscript presents novel perspectives and important findings that contribute to the advancement of EV-based drug delivery systems. The manuscript holds substantial scientific merit and, after addressing minor revisions, will be well-suited for publication given its relevance to current developments in EV-based cancer therapies.
Minor Comments:
- In the section on 2.2. EV Isolation Methodologies, the studies are well described; however, for readers who may be new to EV isolation, the content might be difficult to follow. Adding 2–3 sentences with a general explanation of the principles behind each method would greatly enhance the clarity and accessibility of this section.
- The 2.3.1. Pre-Loading section would benefit from a wider selection of references. Relying on just one source limits the depth of the review, and including more citations would enhance the overall comprehensiveness of the discussion.
- The citation is missing square brackets.
- Line 63: The abbreviation "EVs" should be used here, as it has already been defined earlier in the text.
- Lines 91-96: This paragraph lacks proper citations.
- The names of bacteria should be italicized.
Author Response
Thank you very much for taking the time to review this manuscript. In the attachment, we have labeled the responses more prominently. Please see the attachment.
|
Comments 1: In the section on 2.2. EV Isolation Methodologies, the studies are well described; however, for readers who may be new to EV isolation, the content might be difficult to follow. Adding 2–3 sentences with a general explanation of the principles behind each method would greatly enhance the clarity and accessibility of this section.
|
|
Response 1:Thank you for your advice. We have appended a concise 2-3 sentence explanatory note to each EV isolation method, providing a brief general description of its principles and procedural framework.(Page 3 line 99)
The changes are as follows:
Ultracentrifugation, the current gold-standard method, exploits differential sedi-mentation coefficients of EVs versus other cellular components under centrifugal force. The most widely employed modes of ultracentrifugation include differential centrifugation and density gradient centrifugation. Differential centrifugation involves sequential centrifugation steps at progressively increasing centrifugal forces/speeds to separate heterogeneous particles, including extracellular vesicles (EVs), based on their size and density. Density gradient centrifugation utilizes reagents such as sucrose, deuterium oxide, and/or iodixanol (OptiPrep) to establish a continuous density gradient or discontinuous cushion. This enables refined separation of EVs and their subpopulations by exploiting differences in buoyant density [28]. Despite its standardized protocols, this method necessitates expensive ultracentrifugation equipment, pro-longed processing times (>10 hours), and risks mechanical shear damage to EVs, potentially compromising their bioactivity and downstream functional applications [29]. Size-exclusion chromatography (SEC), also known as gel filtration chromatography, is a chromatographic method that separates particles into distinct fractions during elution based on the relative relationship between the pore size of the stationary phase packing material and the Stokes radius (i.e., the apparent molecular dimensions in solution) of the analytes. SEC is widely employed for EVs isolation from complex biofluids. Although SEC effectively depletes high-abundance proteins like albumin, traditional workflows require multiple fractionation steps with tandem chromatographic columns. To address this, Lin et al. optimized a simplified binary SEC protocol using CL-6B resin, expanded column bed volume (20 mL), and large elution volumes (8 mL), enabling efficient separation of EVs from protein contaminants in just two elution steps [30]. Comparative studies by Yang et al. established that SEC-derived EVs exhibit superior RNA integrity for miRNA/mRNA sequencing and higher concordance with ideal FPKM (Fragments Per Kilobase Million) values in plasma omics analyses, posi-tioning SEC as the optimal strategy for EV-based biomarker discovery [31]. Further advancements include Kapoor et al.'s Size-Exclusion Fast Protein Liquid Chromatography (SE-FPLC), achieving high EVs recovery (88.47%) within 20 minutes while effectively removing lipoprotein contaminants from human/mouse serum and cell-derived EVs [32]. Liu et al. integrated SEC with tangential flow filtration to massively produce functionally distinct sEV subpopulations (S1-sEVs and S2-sEVs) from human umbilical cord mesenchymal stem cells (hUC-MSCs). S1-sEVs (CD9+/HRS+/GPC1+) showed potent immunomodulatory effects, whereas S2-sEVs (CD63+/FLOT1/2+) enhanced angiogenesis and proliferation in disease models [33]. Affinity-based enrichment is a methodology that employs molecules exhibiting high binding affinity to surface markers on EVs, either through facilitating the binding and enhancing the sedimentation coefficient of EVs or by utilizing magnetic beads to achieve EV enrichment. Immunoaffinity capture using magnetic beads functionalized with EV surface markers (CD63/CD9/CD81) achieves high specificity but requires harsh elution conditions (e.g., low pH) that may damage EVs. Brambilla et al. innovated a DNA-directed antibody immobilization system, enhancing capture efficiency via flexible DNA linkers and enabling gentle EV release using DNase I endonuclease [34]. Di et al. developed an automated Fe3O4@TiO2 bead-based platform for simultaneous EV enrichment and miRNA extraction within 30 minutes, outperforming TRIzol and commercial kits in RNA yield [35]. Wang et al. engineered CD63-antibody-conjugated cellulose nanofibers, achieving 86.4% EV capture efficiency through enhanced surface-area interactions [36]. Microfluidic technology enables efficient separation, purification, or active generation of EVs through precisely engineered microscale channels and fluid control mechanisms, leveraging physical effects such as laminar flow, shear stress, or electric fields. BajoSantos et al. designed a cyclic olefin copolymer (COC)-based asymmetrical flow field-flow fractionation (A4F) device for continuous-flow EVs separation, scalable for industrial/clinical production [38]. Loeng et al. fabricated a thiolene polymer-based microfluidic SEC (μSEC) platform with rapid flow-switching capabilities, enabling automated integration with downstream EV analytics [39]. Xin et al. engineered a recyclable boronate organic framework (BOF)-coated microfluidic chip, leveraging ROS-responsive phenylboronic ester crosslinking to achieve size-tunable (10–300 nm pores) EV isolation with reduced cost and technical demands compared to ultracentrifugation [40].
|
|
Comments 2: The 2.3.1. Pre-Loading section would benefit from a wider selection of references. Relying on just one source limits the depth of the review, and including more citations would enhance the overall comprehensiveness of the discussion. |
|
Response 2: We sincerely appreciate your comment. We have added two relevant references in section 2.3.1. Pre-Loading to enhance the overall comprehensiveness of the discussion.(Page 6 line 209 and Page 6 line 215)
The changes are as follows:
54. Zeng H.; Guo S.; Ren X.; Wu Z.; Liu S.; Yao X. Current Strategies for Exosome Cargo Loading and Targeting Delivery. Cells 2023, 12, 1416. https://doi.org/10.3390/cells12101416. 55. Bahadorani M.; Nasiri M.; Dellinger K.; Aravamudhan S.; Zadegan R. Engineering Exosomes for Therapeutic Applications: Decoding Biogenesis, Content Modification, and Cargo Loading Strategies. Int. J. Nanomedicine 2024, 19, 7137–7164. https://doi.org/10.2147/IJN.S464249.
|
|
Comments 3: The citation is missing square brackets. |
|
Response 3: We greatly appreciate you pointing out this issue, and we have added square brackets to all the citations in the article.
|
|
Comments 4: Line 63: The abbreviation "EVs" should be used here, as it has already been defined earlier in the text. |
|
Response 4: We sincerely thank you for pointing out this issue. We have replaced the Extracellular Vesicles in line 63 with EVs.(Page 2 line 70)
The changes are as follows:
The intrinsic biological properties of EVs underpin their efficacy as drug delivery systems.
|
|
Comments 5: Lines 91-96: This paragraph lacks proper citations. |
|
Response 5: We sincerely thank you for pointing out this mistake, and we have added the correct citation for lines 91-96.(Page 3 line 108)
The changes are as follows:
28. Zhang Q.; Jeppesen D.K.; Higginbotham J.N.; Franklin J.L.; Coffey R.J. Comprehensive Isolation of Extracellular Vesicles and Nanoparticles. Nat. Protoc. 2023, 18, 1462–1487, https://doi.org/10.1038/s41596-023-00811-0.
|
|
Comments 6: The names of bacteria should be italicized. |
|
Response 6: We sincerely thank you for pointing out our mistake. We have italicized all bacterial names in the article.(Page 10 line 414)
The changes are as follows:
Bacterial-derived extracellular vesicles (EVs) have emerged as promising plat-forms for cancer drug delivery, leveraging their innate immunogenicity and intrinsic antitumor properties [106]. For instance, EVs from the probiotic Lactobacillus reuteri (REVs) exhibit exceptional stability in the gastrointestinal tract, exerting antitumor effects through modulation of apoptotic signaling pathways. When combined with chemotherapeutic agents, REVs enhance tumor ablation efficacy and induce immuno-genic cell death, demonstrating their dual therapeutic potential [107]. Similarly, Lacto-bacillus paracasei-derived EVs (LpEVs) curb colorectal cancer cell proliferation by sup-pressing HIF-1α-mediated glycolysis, thereby disrupting energy metabolism essential for tumor growth [108]. Parallel findings by Zhang et al. revealed that Lactobacillus plantarum-EVs (LEVs) suppress colorectal cancer progression via SIRT5 downregulation, which modulates p53 desuccinylation to attenuate glycolysis and proliferation [109]. Beyond direct antitumor effects, Bifidobacterium longum-derived EVs reverse carboplatin resistance in ovarian cancer cells by promoting p53 Ser15 phosphorylation, sensitizing tumors to chemotherapy and indirectly enhancing therapeutic outcomes [110]. Despite their therapeutic potential, scalable production of bacterial EVs remains a critical challenge for clinical translation. To address this, Won et al. developed a robust manufacturing process for Escherichia coli-derived outer membrane vesicles (OMVs), which exhibit potent immunostimulatory properties. These OMVs enhance intratumoral infiltration and activation of CD8+ T cells, particularly those expressing high levels of TCF-1 and PD-1, thereby amplifying antigen-specific antitumor immunity. Notably, E. coli OMVs synergize with anti-PD-1 checkpoint inhibitors by promoting the recruitment of stem-like CD8+ T cells into the tumor microenvironment, achieving significant tumor growth suppression in preclinical models [111]. Furthermore, Xiao et al. demonstrated that OMVs reprogram tumor-associated macrophages toward the pro-inflammatory M1 phenotype, stimulating the release of cytokines such as TNF-α and IL-12 to establish an immunologically hostile milieu for cancer progression [112]. Collectively, these studies elucidate the bifunctional capacity of bacterial EVs as both targeted drug carriers and immunomodulators, bridging the gap between microbial biology and precision oncology.
|
Reviewer 2 Report
Comments and Suggestions for Authors
The manuscript titled “Extracellular Vesicle-Based Drug Delivery Systems in Cancer Therapy” by Wu, J.; et al. is a scientific Review work where the authors outlined the most recent advances in the application of extracellular vesicles to fight agains cancer malignancies. The structure is coherent and the authors discussed about the promising avenues in the design of the next-generation of therapies and their challenges. The manuscript is generally well-written and this is a topic of growing interest.
However, it exists some points that need to be addressed (please, see them below detailed point-by-point) to improve the scientific quality of the submitted manuscript paper before this article will be consider for its publication in the International Journal of Molecular Sciences.
1) Introduction. “Cancer remains one of the most formidable threats (…) 9.7 million deaths reported worldwide in 2022 (…) 35 million new cancer caases by 2050” (lines 22-25). Could the authors provide quantitative data insights according to the worldwide global incidence of disability-adjusted life years (DALYs) concerning cancer diseases? This will significantly aid the potential readers to better understand the significance of the topic covered in this Review work.
2) “2.2. EV Isolation Methodologies” (lines 84-148). Here, it should be also mentioned laboratory strategies to obtained giant unillamelar vesicles like the electroformation process, among other alternatives.
3) “2.3.2. Post-Loading” (lines 171-242). A schematic representation will benefit the comprehension by the potential readers.
4) “3. Advantages of Extracellular Vesicle-Based Drug Delivery Systems from Diverse Cellular Origins in Cancer Therapy” (lines 243-448). Here, even if I agree with the content of this section it may be opportune to discuss how magnetic nanomaterials [1] can be coupled with extracellular vesicles to offer more efficient treatments against cancer malignancies [2].
[1] https://doi.org/10.3390/nano13182585
[2] https://doi.org/10.1021/acsami.4c06862
5) “Similarly, Kim et al (…) Error! Reference source not found” (lines 297-301). Some issue related to employment of the bibliography management software has arrived. The authors need to fix it.
6) Finally, in what conditions the extracellular vesicles are not longer viable for their further use in bioclinical research? Some insights should be furnished in this regard
7) “6. Conclusions and Perspectives” (lines 939-970). This section perfectly remarks the most relevant outcomes found by the authors in this field and also the promising future prospectives. It may be advisable to add a brief statement to remark the potential future action lines to pursue the topic covered in this work.
Author Response
Thank you very much for taking the time to review this manuscript. In the attachment, we have labeled the responses more prominently. Please see the attachment.
|
Comments 1: Introduction. “Cancer remains one of the most formidable threats (…) 9.7 million deaths reported worldwide in 2022 (…) 35 million new cancer caases by 2050” (lines 22-25). Could the authors provide quantitative data insights according to the worldwide global incidence of disability-adjusted life years (DALYs) concerning cancer diseases? This will significantly aid the potential readers to better understand the significance of the topic covered in this Review work.
|
|
Response 1: We sincerely appreciate your suggestion. We have added quantitative data on disability adjusted life years (DALYs) incidence of global cancer diseases in the introduction section, further emphasizing the importance of the paper topic.(Page 1 line 22)
The changes are as follows:
Cancer remains one of the most formidable threats to human health and life. According to global statistics from 2022, there were nearly 20 million new cancer cases and 9.7 million cancer-related deaths worldwide[1]. Data from 2021 further revealed that cancer accounted for 14.57% of total deaths and 8.8% of total Disability-Adjusted Life Years (DALYs) globally across both sexes. The Age-Standardized Incidence Rate (ASIR) and Age-Standardized Death Rate (ASDR) were 790.33 and 116.49 per 100,000 population , respectively [2].
|
|
Comments 2: “2.2. EV Isolation Methodologies” (lines 84-148). Here, it should be also mentioned laboratory strategies to obtained giant unillamelar vesicles like the electroformation process, among other alternatives. |
|
Response 2: We sincerely appreciate your suggestion. We have added a laboratory strategy for obtaining giant unillamelar vesicles in "2.2. EV Isolation Methodologies", introducing the current mainstream solutions and related alternative solutions. (Page 5 line 174)
The changes are as follows:
Beyond conventional vesicle preparation methods, the development of giant unilamellar vesicles (GUVs) has garnered increasing attention in recent years. Among existing techniques, electroformation is the most widely utilized approach [49]. This method employs an alternating electric field with specific frequency and intensity to induce asymmetric mechanical stress on phospholipid bilayers by leveraging the electrical properties of lipid headgroups, thereby promoting membrane bending, budding, and eventual vesicle closure [50]. However, traditional electroformation protocols of-ten compromise GUV quality due to residual dried lipid films, which reduce cholesterol concentration within the bilayer. To address this limitation, Boban et al. proposed an optimized electroformation strategy combining rapid solvent exchange, plasma cleaning, and spin-coating techniques to reproducibly generate GUVs from hydrated lipid films. Compared to conventional protocols, this method yields vesicles of comparable size but superior structural integrity [51].In contrast, Waeterschoot et al. introduced a novel approach leveraging fluorinated silica nanoparticles (FNPs) to destabilize lipid-based nanosystems, enabling GUV formation under diverse buffer conditions while preventing leakage of encapsulated components into the oil phase. A simple centrifugation step efficiently releases emulsion-trapped GUVs without requiring destabilizing chemicals. Subsequent evaluations of lipid lateral diffusion and GUV unilamellarity confirmed performance parity with electroformation-derived vesicles [52]. Furthermore, Ernits et al. established a microfluidics-based method for direct synthesis of monodisperse GUVs (~100 μm diameter). Experimental results demonstrated a vesicle half-life of 61 ± 2 hours, coupled with efficient release of small dye molecules without significant membrane disruption. This technique reduces both production time and cost compared to prior methodologies [53].
|
|
Comments 3: “2.3.2. Post-Loading” (lines 171-242). A schematic representation will benefit the comprehension by the potential readers. |
|
Response 3: We sincerely appreciate your suggestion. We have drawn a schematic representation for "2.3.2. Post Loading" to facilitate readers' understanding.(Page 5 line 200)
The schematic representation is as follows:(Please see the attachment)
|
|
Comments 4: “3. Advantages of Extracellular Vesicle-Based Drug Delivery Systems from Diverse Cellular Origins in Cancer Therapy” (lines 243-448). Here, even if I agree with the content of this section it may be opportune to discuss how magnetic nanomaterials [1] can be coupled with extracellular vesicles to offer more efficient treatments against cancer malignancies [2].
[1] https://doi.org/10.3390/nano13182585
[2] https://doi.org/10.1021/acsami.4c06862 |
|
Response 4: We greatly appreciate your suggestion. We first carefully read the two articles you provided and actively cited them. Based on this, we added content on the combination of magnetic nanomaterials and EVs to more effectively treat cancer in "3. Advantages of Extracellular Vesicle Based Drug Delivery Systems from Diverse Cellular Origins in Cancer Therapy".(Page 12 line 518)
The changes are as follows:
Beyond inherent natural advantages, the integration of extracellular vesicles (EVs) with novel nanomaterials holds significant potential to further enhance their functional performance. Among these advancements, the combination of EVs with magnetic nanomaterials has attracted considerable attention. This hybrid system retains the intrinsic biological properties of EVs while incorporating the magnetic characteristics of nanomaterials, offering unique advantages in tumor-targeting specificity and therapeutic efficacy, thereby positioning itself as a promising strategy for optimizing EV-based drug delivery systems [120].For instance, Wang et al. engineered EV-mimetic nanovesicles (NVs) encapsulating iron oxide nanoparticles (IONPs) and β-Lapachone (Lapa). Experimental results demonstrated that the designed NV-IONP-Lapa exhibited enhanced tumor-targeting specificity, facilitated MR imaging, and improved tumor suppression without significant side effects [121].
|
|
Comments 5: “Similarly, Kim et al (…) Error! Reference source not found” (lines 297-301). Some issue related to employment of the bibliography management software has arrived. The authors need to fix it. |
|
Response 5: We sincerely thank you for pointing out the mistake. We have corrected this issue in the latest manuscript.(Page 9 line 367)
|
|
Comments 6: Finally, in what conditions the extracellular vesicles are not longer viable for their further use in bioclinical research? Some insights should be furnished in this regard |
|
Response 6: We sincerely appreciate your suggestion. We have added factors that limit the clinical research of extracellular vesicles in "5. Clinical Advances in Extracellular Vesicles Based Drug Delivery Systems" and explained the conditions under which extracellular vesicles are not suitable for biological clinical research.(Page 27 line 1027)
The changes are as follows:
5.3. Current Limitations in Clinical Research Progress Extracellular vesicles (EVs), an emerging class of drug delivery vehicles, hold immense promise for cancer therapy. However, their clinical translation faces multifaceted challenges spanning technical, ethical, and regulatory domains. From a technical perspective, critical hurdles include the need to optimize EV preparation and purification methods to ensure consistent quality and functionality, as current protocols risk altering their structural integrity and biological activity. Stability during storage and in physiological environments remains another major obstacle, as EVs are prone to enzymatic degradation and oxidation. Most critically, safety concerns necessitate long-term clinical monitoring to evaluate potential side effects, particularly given the risk that EVs may carry tumor-associated molecules, inadvertently promoting tumorigenesis or metastasis [197]. Ethically and legally, safeguarding participant privacy and ensuring informed consent in EV-related studies are non-negotiable priorities. Additionally, the involvement of multiple stakeholders—researchers, patients, and commercial entities—demands transparent frameworks to equitably distribute benefits and protect rights. Regulatory challenges further complicate clinical adoption. Standardized clinical trial protocols are urgently needed to validate the efficacy and safety of EV-based therapies, which remain in early developmental stages. Equally important is establishing rigorous quality control systems to standardize source material selection, production processes, and final product characterization, ensuring reproducibility and safety across batches [198]. In summary, unresolved technical limitations, ethical ambiguities, or regulatory gaps could render EVs unsuitable for clinical research. Future efforts must integrate multidisciplinary strategies to address these barriers, accelerating the translation of EV-based therapies into mainstream clinical practice.
|
|
Comments 7: “6. Conclusions and Perspectives” (lines 939-970). This section perfectly remarks the most relevant outcomes found by the authors in this field and also the promising future prospectives. It may be advisable to add a brief statement to remark the potential future action lines to pursue the topic covered in this work. |
|
Response 7: We sincerely appreciate your suggestion. We have added a short paragraph in "6. Conclusions and Perspectives" about possible future action steps to achieve large-scale application of drug delivery systems using extracellular vesicles.(Page 29 line 1081)
The changes are as follows:
In summary, EVs as natural drug delivery systems, exhibit remarkable advantages in cancer therapy but are accompanied by critical challenges. Future advancements could focus on three key strategies: (1) developing simplified and efficient EV isolation techniques, (2) inventing optimized drug-loading methodologies to enhance cargo capacity, and (3) selecting EV sources with inherent therapeutic potential while integrating advanced materials for functional modifications. Through these approaches, EVs are poised to emerge as pivotal tools in next-generation cancer therapeutics, offering novel solutions to improve treatment efficacy, minimize off-target effects, and advance personalized medicine. |
